# Random Anchors with Low-rank Decorrelated Learning: A Minimalist Pipeline for Class-Incremental Medical Image Classification

**Xinyao Wu[1]\*  Zhe Xu[1,2,3]\*†  Raymond Kai-yu Tong[1]†**

[1]Department of Biomedical Engineering, The Chinese University of Hong Kong
[2]Department of Radiation Oncology, Columbia University Irving Medical Center
[3]Data Science Institute, Columbia University
`{xinyaowu, jackxz}@link.cuhk.edu.hk; kytong@cuhk.edu.hk`

## Abstract

Class-incremental learning (CIL) in medical image-guided diagnosis requires models to preserve knowledge of historical disease classes while adapting to emerging categories. Pre-trained models (PTMs) with well-generalized features provide a strong foundation, yet most PTM-based CIL strategies, such as prompt tuning, task-specific adapters and model mixtures, rely on increasingly complex designs. While effective in general-domain benchmarks, these methods falter in medical imaging, where low inter- and intra-class variability and high inter-domain shifts (from scanners, protocols and institutions) make CIL particularly prone to representation collapse and domain misalignment. Under such conditions, we find that lightweight representation calibration strategies, often dismissed in general-domain CIL for their modest gains, can be remarkably effective for adapting PTMs in medical settings. To this end, we introduce Random Anchors with Low-rank Decorrelated Learning (RA-LDL), a minimalist representation-based framework that combines (a) PTM-based feature extraction with optional ViT-Adapter tuning, (b) feature calibration via frozen Random Anchor projection and a single-session-trained Low-Rank Projection (LRP), and (c) analytical closed-form decorrelated learning. The entire pipeline requires gradient-based optimization only in the first session; subsequent tasks rely solely on efficient analytic classifier updates based on recursively accumulated statistics, making it appealing for efficient deployment. Despite its simplicity, RA-LDL achieves consistent and substantial improvements across both general-domain and medical-specific PTMs, and outperforms recent state-of-the-art methods on four diverse medical imaging datasets. These results highlight that minimalist representation recalibration, rather than complex architectural modifications, can unlock the underexplored potential of PTMs in medical CIL. We hope this work establishes a practical and extensible foundation for future research in class-incremental image-guided diagnosis.

## 1 Introduction

In real-world medical settings, diagnostic models must continually accommodate newly emerging disease categories while preserving expertise on previously learned conditions. This evolving need highlights the importance of class-incremental learning (CIL) for scalable medical image-guided diagnosis. In the general domain, CIL has undergone a paradigm shift with the advent of pre-trained models (PTMs), leveraging the generalized feature representations learned from large-scale datasets to enable more efficient and effective incremental learning. Unlike conventional models trained sequentially from scratch, PTMs inherently possess generalized representations, allowing for improved knowledge retention and faster adaptation to new classes with minimal task-specific tuning.

---

\*Equal contribution.
†Corresponding author.

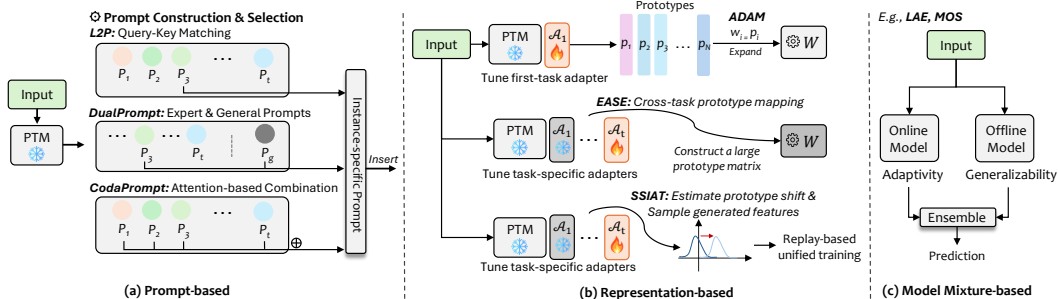

Figure 1: Overview of recent paradigms in the PTM-based CIL. (a) Prompt-based methods (e.g., L2P (Wang et al., 2022c), DualPrompt (Wang et al., 2022b), CodaPrompt (Smith et al., 2023)) dynamically construct and insert instance-specific prompts to guide the frozen PTM. (b) Representation-based methods (e.g., ADAM (Zhou et al., 2024a), EASE (Zhou et al., 2024b), SSIAT (Tan et al., 2024)) focus on adapter tuning and prototype-based classification or calibration. (c) Model Mixture-based methods (e.g., LAE (Gao et al., 2023), MOS (Sun et al., 2025)) combine different models to balance adaptivity and generalizability.

Existing paradigms in the PTM-based CIL can be categorized into three fashions as shown in Fig. 1: prompt-based, representation-based, and model mixture-based methods. While these methods attempt to balance stability and adaptability, they often adopt increasingly complex designs. Prompt-based methods (Wang et al., 2022c;b; Smith et al., 2023) aim to reuse frozen PTMs by introducing learnable context tokens that serve as task-specific prompts. While elegant in theory, they face scalability challenges as the prompt pool grows over time. Managing prompt selection, updating old prompts, and ensuring compatibility between prompts and unseen classes require additional routing mechanisms, leading to increased memory and inference complexity. Moreover, prompts are often sensitive to distribution shifts, making them brittle in cross-domain medical scenarios. Representation-based methods (Zhou et al., 2024b; Tan et al., 2024) build on the strong and conceptually simple baseline SimpleCIL (Zhou et al., 2024a), and typically insert task-specific lightweight modules into the frozen PTM backbone to do the representation decoration. Recent variants (Zhou et al., 2024b; Tan et al., 2024; Sun et al., 2025) further increase complexity by introducing multiple task-specific adapters, requiring task-specific retrieval mechanisms at inference time, reminiscent of prompt-selection overhead. These designs also demand maintaining consistent mappings across evolving feature spaces, often using prototype reconstruction, which can become unstable under domain shift and difficult to scale. Model mixture-based methods (Sun et al., 2025; Gao et al., 2023) require additional mechanisms for effective model ensemble and large memory resources to store all historical models. Yet, this surge in architectural and algorithmic complexity presents a question: *Do these sophisticated methods, successful in general benchmarks, generalize effectively to medical imaging domains?*

Our study (Table 2) reveals that they often do not. Medical imaging poses unique challenges (discussed in Appendix A), including **low inter- and intra-class variability**, where diseases share highly similar visual cues, and **high inter-domain shifts** arising from differences in scanners, protocols, and institutions. Under these conditions, complex mechanisms such as prompt routing, prototype reconstruction, or adapter mixtures tend to collapse or misalign representations. Furthermore, the representation quality of PTMs is uneven. For example, we empirically found that Microsoft's BiomedCLIP (Zhang et al., 2023b) sometimes outperforms Google's ViT-B/16-IN21K (Alexey, 2020) in prototype-based evaluations, yet other medical-specific PTMs such as UniMedCLIP (Khattak et al., 2024) and RAD-DINO (Pérez-García et al., 2025) perform substantially worse, and even BiomedCLIP remains inconsistent across tasks (Table 3). This reveals a critical insight: domain specialization does not guarantee stronger medical representations, whereas well-generalized PTMs from the large-scale natural image domain often provide more reliable foundations. Accordingly, following prior general-domain PTM-based CIL studies (Zhou et al., 2024a; Tan et al., 2024; Sun et al., 2025), we still adopt ViT-B/16-IN21K (Alexey, 2020) as our primary PTM. Here, we wonder: *Can well-generalizable PTMs be effectively adapted to medical CIL through minimalist representation recalibration strategies, rather than increasingly complex architectural and algorithmic designs?*

To this end, we introduce Random Anchors with Low-rank Decorrelated Learning (RA-LDL), a minimalist yet powerful representation-based pipeline for class-incremental medical image classifica-

tion. RA-LDL consists of three lightweight and synergistic steps (Fig. 2), each designed to resolve a core challenge in PTM-based medical CIL: **(a) PTM-based feature extraction with one-time adapter tuning:** Medical images differ substantially from the distributions used in PTM pretraining, raising the challenge of *how to adapt to downstream domains while preserving the generalizability of the foundational representation*. To mitigate this domain gap, we apply a one-time first-session ViT-Adapter update, which improves compatibility with medical images while retaining broad PTM representations. **(b) Feature calibration via frozen random anchors and a low-rank residual:** Under domain shifts, PTM features often lack sufficient discriminability. This raises the key question: *how can we make PTM-derived features more class-separable under domain shifts?* We first apply a frozen Random Anchor (RA) projection to embed features into a randomized higher-dimensional space, enhancing linear separability without additional training. Yet RA alone may be insufficient for severe domain-specific shifts. To address this, we introduce a Low-Rank Projection (LRP), trained only in the first session, that provides a parameter-efficient residual correction. LRP explicitly calibrates residual distortions from domain shift while complementing the frozen RA, yielding features that are both discriminative and robust. **(c) Analytical closed-form decorrelated learning:** Even with calibrated features, *class prototypes in medical CIL often suffer from strong inter-class correlation, especially under domain shifts.* Prototype-based classifiers such as SimpleCIL (Zhou et al., 2024a) demonstrate the generalization strength of PTM features, but their reliance on raw feature prototypes leaves them prone to representation collapse (Table 3). To overcome this, we introduce the analytical learning perspective (Zhuang et al., 2022), framing classifier construction as a closed-form solution rather than iterative backpropagation. Specifically, we estimate classifier weights via ridge regression, whose closed-form solution depends on the feature autocorrelation (Gram) matrix and thus explicitly leverages second-order statistics. This reweighting behaves like a whitening operation: it down-weights dominant and redundant directions, captures intra-class variations, and reduces inter-class correlations. The resulting decorrelated prototypes maintain discriminative power across tasks, align more closely with test features, and improve robustness under domain-shifted medical scenarios. Note that while some components have surfaced in general-domain CIL, to our knowledge, this work provides the first systematic analysis of their synergy and effectiveness in medical CIL. The pipeline needs gradient-based optimization only in the first session; subsequent tasks rely solely on efficient analytic classifier updates based on recursively accumulated statistics, making it attractive for deployment.

Our main contributions are as follows:

- We empirically reveal that while general-domain CIL methods increasingly rely on complex designs, these strategies often fail to generalize in medical imaging, where low inter- and intra-class variability and high inter-domain shifts exacerbate representation collapse. In contrast, simple representation calibration, although yielding only modest gains on general-domain benchmarks, achieves strong and robust performance in medical CIL, highlighting the underappreciated potential of PTMs.

- We introduce RA-LDL, a minimalist pipeline for class-incremental medical image classification. Built on three lightweight representation-based steps, it combines theoretical analysis and empirical validation to demonstrate improvements in class separability, feature adaptability, and generalizability within a single training session.

- We evaluate RA-LDL on four diverse medical CIL benchmarks, demonstrating substantial and consistent gains over state-of-the-art methods. The framework approaches the joint-training upper bounds while requiring gradient-based optimization only in the first session. Moreover, RA-LDL achieves consistent improvements with both general-domain and medical-specific PTMs, further confirming its robustness across different backbone choices.

## 2 RELATED WORK

**Class-incremental Learning.** Conventional CIL methods rely on three strategies: replaying prior data (Castro et al., 2018; Hou et al., 2019; Rebuffi et al., 2017), regularizing adversarial parameter updates (Li & Hoiem, 2017; Kirkpatrick et al., 2017; Wang et al., 2022a), and using adaptive modules to update models dynamically (Abati et al., 2020; Yoon et al., 2017). However, these methods generally require models training from scratch and necessitate extensive parameter tuning,

which limits their scalability especially in practical deployment. Recent advances in PTMs have revitalized CIL research by leveraging their generalized feature representations (Wang et al., 2022c;b; Smith et al., 2023; Zhou et al., 2024a; Tan et al., 2024; Gao et al., 2023; Zhou et al., 2024b; Tan et al., 2024; Sun et al., 2025). However, as discussed above, the effectiveness of these methods often hinges on architectural and algorithmic complexity, which hinders scalability and struggles to generalize consistently in cross-domain tasks such as medical imaging. Interestingly, recent evidence (Zhuang et al., 2024; McDonnell et al., 2024) suggests that even lightweight feature calibration mechanisms—for instance, mapping features into suitably structured spaces—can yield strong incremental performance with minimal adaptation. While more sophisticated designs may outperform these strategies, they impressively reveal the untapped potential of PTM-extracted features for incremental learning. Remarkably, in medical imaging, where inter- and intra-class variability is low and inter-domain shifts are pronounced, we found that simple yet well-crafted feature calibration can achieve competitive CIL performance, in some cases approaching the joint-training upper bound.

**Class-incremental Learning in Medical Image Analysis.** Previous work in medical CIL has primarily focused on training models from scratch, often relying on replay-based techniques or parameter-inefficient full-model regularization (Perkonigg et al., 2021; Byun et al., 2023; Bai et al., 2023; Chee et al., 2023; Wu et al., 2023; Li & Jha, 2023; Yeganeh et al., 2023; Sadafi et al., 2023; Wu et al., 2025). With the emergence of PTMs offering strong generalization, the challenge should shift to effectively balancing plasticity and stability during continual adaptation. Yet, PTM-based CIL in the medical domain remains underexplored, and existing medical PTMs still tend to be modality-specific with poor transferability across medical sub-domains. This work presents an early exploration of PTM-based medical CIL, showing that a minimalist representation-based incremental learning pipeline can achieve excitingly strong performance.

## 3 PRELIMINARIES

### 3.1 PROBLEM SETUP

Class-incremental learning (CIL) aims to learn from a continual data stream where new classes are incrementally introduced. Formally, a training set that belongs to the incremental session $t$ can be represented as $\mathcal{D}^t = \{(x_{i,t}, y_{i,t})\}_{i=1}^{n_t}$, where $t \in \{1, 2, \ldots, T\}$ represents the session index among $T$ total session, and each session contains $n_t$ instances. Mark that each session $t$ contains a unique set of classes $Y_t$, with no overlap between sessions: $Y_t \cap Y_{t'} = \varnothing$ for $t \neq t'$. The goal of CIL is to construct a model $f(\mathbf{x}) : X \to \mathcal{Y}_t$, where $\mathcal{Y}_t = Y_1 \cup \cdots Y_t$, that can learn new classes incrementally without forgetting previously learned ones. Following prior replay-free PTM-based CIL studies (Tan et al., 2024; Zhang et al., 2023a; Zhou et al., 2024a;b), we consider a pre-trained Vision Transformer (ViT) model available for initializing $f(\mathbf{x})$. We then decompose it into a feature extraction backbone $\mathcal{F}_{\theta_{\text{bne}}}$ and a linear classification layer $f_{\theta_{\text{cls}}}$. The backbone $\mathcal{F}_{\theta_{\text{bne}}}$ extracts features from the input images, serving as a feature embedding function $\phi(\cdot)$ mapping $\mathbb{R}^D \to \mathbb{R}^{d_0}$, while the classifier layer $f_{\theta_{\text{cls}}}$, represented by a weight matrix $W \in \mathbb{R}^{d_0 \times |Y_t|}$, projects the feature embeddings to class predictions. The model can then be formularized as $f(\mathbf{x}) = W^\top \phi(\mathbf{x})$, where $W = [\mathbf{w}_1, \mathbf{w}_2, \cdots, \mathbf{w}_j]$, with $\mathbf{w}_j$ denoting the classifier weights for class $j$. We treat the embedded `[CLS]` token as $\phi(\mathbf{x})$ for ViT.

### 3.2 ADAPTIVITY AND GENERALIZABILITY

In PTM-based CIL, a key challenge lies in balancing adaptivity (the model's ability to learn new classes efficiently) and generalizability (the capacity of PTMs to transfer to downstream tasks without learning). Parameter-efficient tuning (PET) methods, such as Scaling and Shifting Fine-tuning (SSF), ViT-adapters and Visual Prompt Tuning (VPT), are widely adopted to enhance adaptivity by modifying and tuning only a small subset of parameters in the PTMs (details in appendix B). During training, the majority of pretrained weights remain frozen, with only the PET components and classifier head being optimized. The set of trainable parameters is defined as: $\Theta = \theta_{W_{\text{PET}}} \cup \theta_W$. While these techniques enable efficient adaptation to downstream domains, they often compromise generalizability, leading to forgetting of prior knowledge. To address this, recent studies introduce increasingly complicated strategies upon the SimpleCIL baseline (Zhou et al., 2024a), such as replay-based unified training (Tan et al., 2024) and cross-task prototype mapping (Zhou et al., 2024b), to better balance the trade-off. However, such complexity could hinder scalability and interpretability,

Table 1: Overview of medical image classification datasets.

| Dataset | Classes | Training set | Test set | Task Num. | Size |
|---|---|---|---|---|---|
| Covid (CT&X-ray) (Wang et al., 2023) | 11 | 3,939 | 1,072 | 6 | $224 \times 224$ |
| Blood (Acevedo et al., 2019) | 8 | 11,965 | 5,127 | 4 | $360 \times 363$ |
| Skin8 (Tschandl et al., 2018) | 8 | 3,555 | 705 | 4 | $[600, 1024]$ |
| MedMNIST-Sub (Yang et al., 2023) | 36 | 302,002 | 75,659 | 4 | $28 \times 28$ |

and has empirically shown inconsistent or even controversial performance in medical CIL. This motivates us to introduce a minimalist pipeline.

### 3.3 MATERIALS AND IMPLEMENTATION

Since we will progressively combine theoretical analysis and empirical evidence to construct the RA-LDL pipeline step by step, we first introduce the datasets and implementation protocol.

**Datasets.** We involve four medical image classification datasets, as summarized in Table 1. COVID (CT & X-ray) (Wang et al., 2023) contains both CT and X-ray scans for respiratory disease diagnosis. Blood (Acevedo et al., 2019) comprises normal peripheral blood cell images obtained from blood smear slides. Skin8 (Tschandl et al., 2018) focuses on skin lesion classification from dermatoscopic images and presents high class imbalance. MedMNISTv2 (Yang et al., 2021; 2023) provides a standardized benchmark with 12 2D and 6 3D biomedical datasets covering multi-class, multi-label, and ordinal regression tasks. We follow (Zhang et al., 2023c) and adopt four classification datasets, BloodMNIST, OrganAMNIST, PathMNIST and TissueMNIST, referred to as MedMNIST-Sub. We use the same data splits as in (Bayasi et al., 2024; Zhang et al., 2023c), and each image is resized to $224 \times 224$ pixels during training. As supplementary validation, we also evaluate on general-domain benchmarks, including CIFAR100 (Krizhevsky et al., 2009), CUB200 (Wah et al., 2011), ImageNet-A (IN-A) (Hendrycks et al., 2021b), and ImageNet-R (IN-R) (Hendrycks et al., 2021a).

**Implementation.** Our experiment is implemented on PyTorch on a NVIDIA A100 GPU. We primarily utilize ViT-B/16-IN21K (Alexey, 2020) as the PTM, since general-domain PTMs remain the mainstream choice whereas medical-specific PTMs are still relatively underexplored. However, in Sec. 5.2, we analyze the sensitivity and comparative advantages of RA-LDL when applied to both general-domain and medical-specific PTMs. All experiments use a batch size of 48, with 20 training epochs for the initial session and 15 epochs for each subsequent session. We employ SGD with momentum and use a cosine-annealed learning rate starting at 0.01. We apply random flipping and rotation for weak data augmentation to enhance generalization. Following (Tan et al., 2024; Zhang et al., 2023a; Zhou et al., 2024a;b), we report the final-session accuracy $Acc_{\text{Last}}$ and the average accuracy across all sessions, defined as $Acc_{\text{Avg}} = \frac{1}{T} \sum_{t=1}^{T} Acc_t$. As the ultimate goal of CIL is to maintain knowledge after all tasks, $Acc_{\text{Last}}$ is the most critical metric, while $Acc_{\text{Avg}}$ reflects stability throughout training. All methods are reproduced using the same seed for fair comparison. Code is available at `https://github.com/CUHK-BMEAI/RA-LDL`.

## 4 METHODOLOGY

Our proposed RA-LDL pipeline operates in three steps, as shown in Fig. 2. In step (a), we extract features from a frozen pre-trained backbone. A lightweight ViT adapter is optionally inserted to support task-specific adaptation, which is only trainable during the first task ($t = 1$) and kept frozen in subsequent tasks ($t > 1$). In step (b), we adopt a frozen random anchor and a learnable low-rank projection (LRP) to complementarily calibrate representations. In step (c), we perform the analytical closed-form decorrelated classification based on ridge regression. This pipeline maintains minimal architectural and training overhead while enabling a controllable balance between downstream adaptability and PTM generalizability in medical continual learning.

### 4.1 FIRST-STAGE ADAPTATION AND ANALYTICAL DECORRELATED CLASSIFIER

We build our pipeline on top of an optional ViT-adapter-based first-stage adaptation, which has been observed to provide stable performance, together with the analytical decorrelated classifier

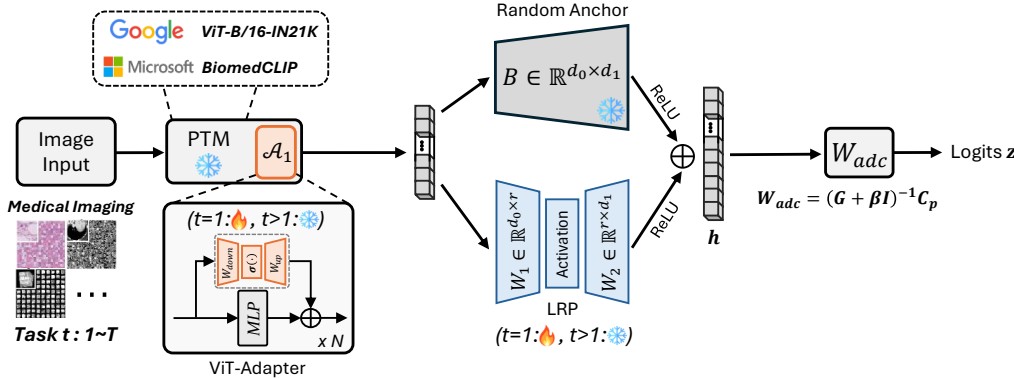

Figure 2: Overview of our minimalist RA-LDL framework. The pipeline requires only a single training session and consists of three lightweight components: (a) PTM-based Feature Extraction with optional ViT-Adapter-based adaptation. (b) Random Anchor & LRP to calibrate representations. (c) Analytical Decorrelated Classifier based on ridge regression.

(Zhuang et al., 2022). Details of alternative PET variants and classifier formulations are provided in Appendix B and Appendix C.

A common choice in CIL is the class-prototype (CP) classifier (Zhou et al., 2024a), which computes class means and applies the nearest-class-mean (NCM) principle (Mensink et al., 2013). This approach implicitly assumes that class embeddings are isotropic and well separated. In PTM-based medical CIL, however, prototypes often show strong inter-class correlation due to domain shifts, resulting in representation collapse and reduced discriminability. Addressing this issue requires a classifier that not only estimates prototypes but also explicitly reduces correlations between them. Motivated by this, we adopt the analytical learning perspective (Zhuang et al., 2022), where classifier construction is formulated as a ridge regression problem instead of relying on iterative optimization. Specifically, we assume that we have projected feature $h(\mathbf{x}) \in \mathbb{R}^{d_1}$ for each input $\mathbf{x}$ in general. To capture second-order relationships and variance structure, we compute the feature autocorrelation (Gram) matrix across sessions 1 to $t$: $G = \sum_t \sum_{n=1}^{N_t} h_{t,n} h_{t,n}^\top \in \mathbb{R}^{d_1 \times d_1}$, and the class-accumulated feature matrix: $C_p = \sum_t \sum_{n=1}^{N_t} h_{t,n} y_{t,n}^\top \in \mathbb{R}^{d_1 \times |Y_t|}$. The analytical classifier weights are obtained by solving the ridge regression problem:

$$\arg \min_{W_{adc}} \|Y - HW_{adc}\|_F^2 + \beta \|W_{adc}\|_F^2 , \tag{1}$$

where $H \in \mathbb{R}^{N \times d_1}$ is the stacked matrix of all projected sample features; $Y \in \mathbb{R}^{N \times |Y_t|}$ contains the corresponding one-hot labels; $N = \sum_{t=1}^T N_t$ is the total number of samples accumulated up to the current task; $\|\cdot\|_F$ is the Frobenius norm; and $\beta$ is the ridge regularization parameter selected by cross-validation-based optimization. Differentiating the ridge regression objective in Eq. 1 yields the closed-form solution (Appendix D), which can be compactly represented using the accumulated feature statistics from sessions 1 to $t$:

$$\hat{W}_{adc} = (G + \beta I)^{-1} C_p \in \mathbb{R}^{d_1 \times |Y_t|}, \tag{2}$$

where $I$ is the identity matrix. The logits for projected feature $h$ can be obtained as $z = h\hat{W}_{adc} = h(G + \beta I)^{-1} C_p \in \mathbb{R}^{|Y_t|}$. This formulation shows that $\hat{W}_{adc}$ corresponds to a set of analytically derived and decorrelated prototypes. The inverse Gram matrix $(G + \beta I)^{-1}$ reweights eigendirections of the feature space, acting similarly to whitening: it suppresses dominant shared components (the source of prototype correlation), captures intra-class variations, and reduces inter-class correlations. This yields prototypes that are more discriminative across tasks and more robust to medical-specific challenges such as low inter- and intra-class variability and high inter-domain shifts. Appendix E discusses the privacy-preserving nature of maintaining only aggregated statistics. Formal proofs of the decorrelation property are provided in Appendix F, together with empirical results showing that the analytical decorrelated classifier consistently outperforms the standard prototype-based classifier.

## 4.2 RANDOM ANCHORS

Recent studies (Zhuang et al., 2024; McDonnell et al., 2024) have shown that nonlinear transformation can improve linear separability among expected class representatives. As shown in Fig. 2 (b), we adopt a lightweight random anchor to transform frozen features while preserving their intrinsic structure. Specifically, we project features into higher-dimensional spaces in a characteristics-preserving manner. We define a frozen random matrix $\mathbf{B} \in \mathbb{R}^{d_0 \times d_1}$, randomly initialized from a Gaussian distribution $\mathcal{N}(0, \sigma^2)$, and apply the transformation:

$$h_{RA}(\mathbf{x}) = \text{ReLU}(\mathbf{B}^\top \phi(\mathbf{x})), \tag{3}$$

where $d_0$ is the feature embedding dimension and $d_1$ is the projected dimension. $\phi(\mathbf{x})$ is the extracted feature from the frozen pretrained model, and $h_{RA}(\mathbf{x})$ is the projected feature. The matrix $\mathbf{B}$ is initialized once and frozen throughout all incremental learning stages. This projection introduces minimal perturbation to the embedding space while improving class separability.

**Proof of Feature Characteristics Preservation and Enhancement.** In Appendix H, we formally prove that random anchor projections guided by the Johnson-Lindenstrauss lemma effectively preserve original feature characteristics and, when used to increase dimensionality, can potentially improve the structural and statistical properties of features extracted from pre-trained models.

**Impact of Different Random Matrix.** Given the diversity of weight initialization techniques, we conducted a study to determine the optimal strategy for initializing the random matrix. Our evaluation includes widely-adopted methods, including Kaiming uniform/normal and Xavier uniform/normal distributions. As shown in Appendix G.1, the standard Gaussian initialization demonstrates the most stable and consistently strong performance. While other methods yield competitive results in some datasets, they exhibit greater variance across tasks and datasets. In contrast, the Gaussian initialization reliably provides high performance without requiring fine-tuned hyperparameters, aligning well with our pipeline in a minimalist perspective.

**Impact of Projection Dimension.** To better evaluate the randomness introduced by the frozen random anchor and how it helps improve class prototype separability, we analyze how the projection dimension $d_1$ influences performance. As shown in Appendix G.2, increasing $d_1$ generally improves linear separability by expanding the feature space, and thus yields some performance gains. However, the effect is not uniform across datasets: while some benefit steadily from larger projections, others show fluctuations or diminishing returns. This variability reflects the fact that RA, though effective at preserving global geometry and enhancing separability, does not explicitly account for domain-specific distortions. These observations motivate the need for a complementary adaptation branch: rather than relying solely on random projection, we introduce a parameter-efficient Low-Rank Projection (LRP) to provide a residual correction. For all subsequent experiments, we set $d_1 = 5d_0$, which we find provides a sufficiently large feature space.

## 4.3 LOW-RANK PROJECTION LAYER

Random Anchors (RA) effectively preserve global feature geometry and improve separability through higher-dimensional embedding. However, RA alone does not explicitly address domain-specific variations, which are particularly pronounced in medical imaging where a substantial distribution gap between pre-training and downstream domains often exists. In this regard, we introduce a *Low-Rank Projection* (LRP) trained in the first session. This module provides a residual parameter-efficient correction to the frozen RA pathway. We define the composite output as:

$$h(\mathbf{x}) = h_{RA}(\mathbf{x}) + h_{LRP}(\mathbf{x}), \tag{4}$$

where $h_{RA}(\mathbf{x}) = \text{ReLU}(\mathbf{B}^\top \phi(\mathbf{x}))$ is the frozen RA projection and $h_{LRP}(\mathbf{x}) = \text{ReLU}(\text{Act}(\phi(\mathbf{x})\mathbf{W}_1) \cdot \mathbf{W}_2)$ is the trainable low-rank residual. Here, $\mathbf{W}_1 \in \mathbb{R}^{d_0 \times r}$, $\mathbf{W}_2 \in \mathbb{R}^{r \times d_1}$, and $r \ll \min(d_0, d_1)$. The activation function $\text{Act}(\cdot)$ is set to GELU (Hendrycks & Gimpel, 2016). This design aims to preserve the core structure learned by the PTM through RA, while the LRP corrects domain-specific distortions in a compact and regularized fashion. The residual structure enables the model to bridge the domain gap with minimal parameters and computation.

**Proof of LRP residual-induced Variance Reduction and Margin Enlargement.** We further provide theoretical analysis in Appendix I, proving that the LRP residual reduces intra-class variance

while enlarging inter-class margins. With the low-rank constraint ($r \ll \min(d_0, d_1)$), the parameter count of the residual correction ($(d_0 + d_1)\, r$) is significantly smaller than the full linear layer ($d_0 d_1$), remaining parameter-efficient to alleviate overfitting.

**Impact of Rank $r$.** Appendix G.3 presents a sensitivity analysis examining how varying the rank $r$ affects the performance of our low-rank projection module. The rank determines the trade-off between model capacity and parameter efficiency by controlling the number of trainable parameters and the dimensionality of the learned correction. As shown in Table 8, increasing the rank from 64 to 256 yields only marginal improvements in accuracy. This indicates that a moderate rank (e.g., $r = 64$) is sufficient to capture the dominant domain-specific variations required for effective adaptation, while preserving the parameter efficiency and regularization benefits of the low-rank design.

## 5 EXPERIMENTAL ANALYSIS

### 5.1 COMPARISON WITH STATE-OF-THE-ART METHODS

After progressively designing and validating each component, we integrate them into a unified three-step RA-LDL pipeline. Table 2 compares our approach with recent PTM-based CIL methods on four challenging medical imaging benchmarks. We also include traditional CIL methods (Rebuffi et al., 2017; Wang et al., 2022a; Yan et al., 2021); while they perform competitively on the Blood dataset, their reliance on data replay raises privacy concerns in clinical scenarios. Excitingly, RA-LDL achieves striking performance using minimalist representation recalibration strategies, highlighting its effectiveness in handling cross-domain tasks with heterogeneous data without using complicated architectures or heavy loss regularization. Fig. 3 further illustrates the performance trajectories across incremental sessions, where RA-LDL sustains more robust accuracy as class numbers grow, while competing methods show sharp declines, particularly on Skin8 and COVID. Appendix G.4 shows that RA-LDL achieves comparably competitive performance on general-domain benchmarks, validating the generalizability of our minimalist pipeline beyond the medical domain. Appendix G.5 further shows that RA-LDL achieves the best overall performance on our cross-dataset continual learning benchmark, surpassing recent PTM-based CIL methods in both average and final accuracy. The last four rows of Table 2 present the ablation results where we remove first-stage adaptation and LRP residual. The performance exhibits mixed outcomes. This highlights the insights: the first-stage adaptation is optional, whereas the combination of the random anchor and the LRP residual plays a complementary and essential role in balancing generalizability and adaptivity. Together, they form a minimalist yet effective pipeline for recalibrating representations in incremental tasks, requiring minimal training overhead and offering excellent scalability without complex architectural designs.

Table 2: Experimental results compared to recent PTM-based CIL methods on four medical classification benchmarks. "†" denotes reliance on replayed data. The best and second-best results are marked in **red** and *blue*, respectively.

| Method | MedMNIST-Sub | | Skin8 | | COVID (CT&X-rays) | | Blood | |
|---|---|---|---|---|---|---|---|---|
| | $Acc_{Avg}$ (%) | $Acc_{Last}$ (%) | $Acc_{Avg}$ (%) | $Acc_{Last}$ (%) | $Acc_{Avg}$ (%) | $Acc_{Last}$ (%) | $Acc_{Avg}$ (%) | $Acc_{Last}$ (%) |
| Joint-training | - | 73.61 | - | 67.73 | - | 92.43 | - | 99.61 |
| Finetune | 29.18 | 5.66 | 39.62 | 17.87 | 26.55 | 10.54 | 37.47 | 14.54 |
| FOSTER† (Wang et al., 2022a) | 58.24 | 31.46 | 55.00 | 39.01 | 71.65 | 55.50 | 87.70 | 90.24 |
| iCaRL† (Rebuffi et al., 2017) | 68.23 | 38.44 | 58.45 | 39.57 | 70.10 | 65.21 | 81.41 | 81.57 |
| DER† (Yan et al., 2021) | 69.97 | 42.11 | 48.57 | 23.55 | 31.40 | 27.05 | 86.48 | 87.10 |
| ACL† (Zhang et al., 2023c) | 81.33 | 65.40 | 58.30 | 57.66 | 85.13 | 69.23 | 76.09 | 82.33 |
| L2P (Wang et al., 2022c) | 56.24 | 28.96 | 55.39 | 35.89 | 43.48 | 20.06 | 86.46 | 76.15 |
| DualPrompt (Wang et al., 2022b) | 54.92 | 26.31 | 52.32 | 27.38 | 43.62 | 19.68 | 76.62 | 66.27 |
| CodaPrompt (Smith et al., 2023) | 61.12 | 28.65 | 50.17 | 32.48 | 36.67 | 20.88 | 72.64 | 61.57 |
| LAE (Gao et al., 2023) | 48.46 | 18.39 | 49.52 | 24.40 | 46.26 | 21.83 | 55.51 | 33.94 |
| SLCA (Zhang et al., 2023a) | 56.39 | 44.42 | 58.91 | 40.71 | 63.05 | 60.82 | 90.23 | 82.29 |
| SimpleCIL (Zhou et al., 2024a) | 68.07 | 50.63 | 56.61 | 38.30 | 75.84 | 57.37 | 83.85 | 79.79 |
| ADAM-Adapter (Zhou et al., 2024a) | 70.97 | 53.11 | 59.82 | 41.84 | 79.19 | 61.10 | 88.09 | 83.52 |
| SSIAT (Tan et al., 2024) | 59.43 | 25.79 | 60.46 | 41.99 | 72.00 | 60.17 | 86.00 | 84.63 |
| EASE (Zhou et al., 2024b) | 65.11 | 39.26 | 60.40 | 40.43 | 78.47 | 59.98 | 68.85 | 67.60 |
| MOS (Sun et al., 2025) | 74.59 | 51.80 | 68.54 | 51.77 | 89.96 | 80.60 | 92.53 | 90.18 |
| Original ViT w/ B (**RA-DL**) | 83.82 | 69.32 | *75.35* | 61.26 | 94.87 | 86.94 | 97.90 | 97.18 |
| Adapted ViT w/ B (**RA-DL**) | **85.23** | 70.00 | 72.83 | *61.42* | *96.28* | *89.55* | 97.97 | 97.45 |
| Original ViT w/ B&LRP (**RA-LDL**) | 84.30 | *70.29* | 75.33 | 60.85 | **96.47** | **90.30** | *98.06* | *97.74* |
| Adapted ViT w/ B&LRP (**RA-LDL**) | *84.79* | **70.60** | **75.66** | **62.49** | 95.98 | 88.04 | **98.07** | **97.76** |

### 5.2 SENSITIVITY ANALYSIS

**Robustness to General-domain and Medical-specific PTMs.** While our main experiments primarily adopt general-domain PTMs such as ViT-B/16-IN21K, an important question is whether RA-LDL

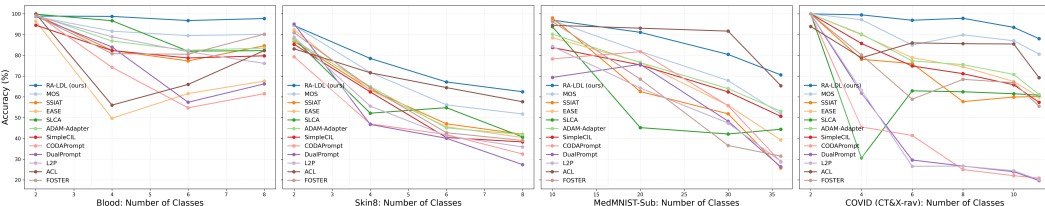

Figure 3: The performance curves across learning sessions on four medical CIL benchmarks.

also generalizes to medical-specific PTMs. To this end, as shown in Table 3, we further evaluate RA-LDL on recent medical PTMs, including BiomedCLIP (Zhang et al., 2023b), UniMedCLIP (Khattak et al., 2024), and RAD-DINO (Pérez-García et al., 2025). Interestingly, we find that features from these medical PTMs do not consistently outperform those from general-domain PTMs, suggesting that domain specialization does not guarantee stronger representations across diverse medical subsets. Nevertheless, RA-LDL consistently yields stable improvements across both general-domain and medical-specific PTMs, confirming its robustness to backbone choice. Moreover, we observe complementary strengths: when applied to medical PTMs, RA-LDL can better leverage domain-specific priors; when applied to general-domain PTMs, its lightweight recalibration effectively mitigates domain shifts. These results suggest that RA-LDL can serve as a unifying framework to unlock the potential of both general-domain and medical-specific PTMs for medical CIL. It is important to note, however, that all strategies rest on the assumed strong generalizability of PTMs: efficient adaptation alone cannot fully compensate for weak representations. Ultimately, the achievable ceiling remains bounded by the quality of the underlying feature extractor. We therefore believe that advances in PTM quality and CIL strategies should evolve together, jointly driving progress in medical continual learning.

Table 3: Performance comparison using different PTMs (medical-specific vs. general-domain).

| Method | PTM | MedMNIST-Sub | | Skin8 | | COVID (CT&X-rays) | | Blood | |
|---|---|---|---|---|---|---|---|---|---|
| | | $Acc_{Avg}$ (%) | $Acc_{Last}$ (%) | $Acc_{Avg}$ (%) | $Acc_{Last}$ (%) | $Acc_{Avg}$ (%) | $Acc_{Last}$ (%) | $Acc_{Avg}$ (%) | $Acc_{Last}$ (%) |
| SimpleCIL (Zhou et al., 2024a) | UniMed-CLIP | 35.10 | 28.93 | 35.28 | 18.72 | 41.44 | 23.60 | 49.47 | 35.64 |
| SimpleCIL (Zhou et al., 2024a) | RAD-DINO | 48.62 | 38.01 | 34.82 | 21.56 | 57.09 | 46.60 | 43.14 | 25.29 |
| SimpleCIL (Zhou et al., 2024a) | BiomedCLIP | 69.86 | 53.21 | 57.43 | 40.99 | 83.51 | 65.30 | 85.86 | 85.17 |
| SimpleCIL (Zhou et al., 2024a) | ViT-B/16-IN21K | 68.07 | 50.63 | 56.61 | 38.30 | 75.84 | 57.37 | 83.85 | 79.79 |
| ADAM-Adapter (Zhou et al., 2024a) | UniMed-CLIP | 48.73 | 33.27 | 33.49 | 21.56 | 42.68 | 30.41 | 41.34 | 22.06 |
| ADAM-Adapter (Zhou et al., 2024a) | RAD-DINO | 49.91 | 38.06 | 35.40 | 21.99 | 55.51 | 39.46 | 42.50 | 24.55 |
| ADAM-Adapter (Zhou et al., 2024a) | BiomedCLIP | 70.88 | 53.48 | 54.79 | 38.87 | 83.38 | 65.21 | 79.12 | 78.74 |
| ADAM-Adapter (Zhou et al., 2024a) | ViT-B/16-IN21K | 70.97 | 53.11 | 59.82 | 41.84 | 79.19 | 61.10 | 88.09 | 83.52 |
| MOS (Sun et al., 2025) | UniMed-CLIP | 70.39 | 57.00 | 52.39 | 35.89 | 79.86 | 69.87 | 86.88 | 86.30 |
| MOS (Sun et al., 2025) | RAD-DINO | 22.95 | 8.23 | 24.09 | 5.53 | 21.28 | 11.01 | 31.68 | 9.08 |
| MOS (Sun et al., 2025) | BiomedCLIP | 78.67 | 63.96 | 67.48 | 51.91 | 90.62 | 73.97 | 90.33 | 89.58 |
| MOS (Sun et al., 2025) | ViT-B/16-IN21K | 74.59 | 51.80 | 68.54 | 51.77 | 89.96 | 80.60 | 92.53 | 90.18 |
| RA-LDL | UniMed-CLIP | 75.03 | 59.67 | 49.32 | 35.18 | 73.40 | 63.25 | 84.54 | 81.64 |
| RA-LDL | RAD-DINO | 75.49 | 61.46 | 53.64 | 38.01 | 79.53 | 67.54 | 68.02 | 52.79 |
| RA-LDL | BiomedCLIP | *84.33* | *70.04* | *73.71* | *61.70* | *94.42* | *87.78* | *97.49* | *96.63* |
| RA-LDL | ViT-B/16-IN21K | **84.79** | **70.60** | **75.66** | **62.49** | **95.98** | **88.04** | **98.07** | **97.76** |

*Note:* SimpleCIL is a feature-quality check baseline presented in (Zhou et al., 2024a), using a frozen PTM with a nearest-class-mean classifier.

**Sensitivity to Task Order.** We further evaluate RA-LDL under a reversed task sequence (Table 4). On COVID, Skin8, and Blood, both $Acc_{Avg}$ and $Acc_{Last}$ vary by less than 1.5%, confirming strong order robustness. On MedMNIST-Sub, $Acc_{Avg}$ shows a moderate drop (~11%), reflecting the high heterogeneity of tasks, yet importantly $Acc_{Last}$ still improves by ~3%. Since the final-session performance is the most critical measure in continual learning, these results demonstrate that RA-LDL remains competitive even under unfavorable task orders. Overall, RA-LDL exhibits desirable robustness to task sequencing, a key property for realistic scenarios where task order is uncontrollable.

Table 4: The performance of our RA-LDL using a reverse task order.

| Method | COVID (CT&X-rays) | | Skin8 | | MedMNIST-Sub | | Blood | |
|---|---|---|---|---|---|---|---|---|
| | $Acc_{Avg}$ (%) | $Acc_{Last}$ (%) | $Acc_{Avg}$ (%) | $Acc_{Last}$ (%) | $Acc_{Avg}$ (%) | $Acc_{Last}$ (%) | $Acc_{Avg}$ (%) | $Acc_{Last}$ (%) |
| RA-LDL | 95.98 | 88.04 | 75.66 | 62.49 | 84.79 | 70.60 | 98.07 | 97.76 |
| RA-LDL (reverse) | 94.58 | 86.76 | 74.70 | 62.20 | 73.77 | 73.51 | 99.04 | 97.37 |

**Robustness to the Number of Tasks.** We further assess the impact of task granularity by increasing the number of tasks on MedMNIST-Sub from 4 to 8, comparing RA-LDL with representative baselines. As shown in Appendix G.6, RA-LDL maintains stable performance: $Acc_{Avg}$ only changes from 84.79% (4 tasks) to 86.71% (8 tasks), and $Acc_{Last}$ from 70.60% to 70.80%. This demonstrates RA-LDL's scalability and strong long-term continual learning potential.

# 6 CONCLUSION

In this paper, we introduced RA-LDL, a minimalist pipeline for class-incremental learning (CIL) in medical image classification that serves as a practical alternative to existing complex PTM-based designs. RA-LDL applies a frozen random anchor projection and a single-session-trained low-rank projection to calibrate extracted features, followed by an analytical ridge-regression-based classifier that produces decorrelated prototypes and maintains discriminative power over time. Without relying on medical-specific pretraining or repeated task-specific tuning, RA-LDL delivers consistently strong performance across four diverse medical datasets. Extensive experiments show that it not only surpasses recent state-of-the-art methods but also reveals the underexplored potential of simple representation refinement strategies in medical CIL. Looking ahead, RA-LDL opens promising directions for future work. Its minimalist design positions it as a lightweight adapter for continual updates in emerging foundation models, while extensions beyond classification to tasks such as segmentation and detection can further demonstrate its scalability and impact.

## ACKNOWLEDGMENTS

This research was partly supported by Research Impact Fund (R5039-23) from Research Grants Council of Hong Kong.

## ETHICS STATEMENT

Our study does not involve human subjects, personally identifiable information, or sensitive clinical data. All datasets used are publicly available medical imaging benchmarks (e.g., MedMNIST, Skin8, COVID-CT/X-ray, and Blood), which have been properly anonymized by their providers. We have carefully considered potential risks and believe that our work poses no foreseeable harm to patients or institutions. This research strictly adheres to the ICLR Code of Ethics.

## REPRODUCIBILITY STATEMENT

All implementation details, including model architectures, training hyperparameters, and data pre-processing steps, are described in the main paper and Appendix. Additional ablation studies and sensitivity analyses are provided in the supplementary material, and complete proofs of theoretical results are included in the Appendix. The source code, along with documentation and scripts to reproduce our experiments, will be released upon paper acceptance.

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

# Appendix

## A    WHY MEDICAL CIL DIFFERS FROM GENERAL-DOMAIN CIL

A critical distinction between continual learning in medical imaging and general-domain tasks lies in the intrinsic data characteristics. Two properties in particular make medical CIL more challenging:

**(1) Low inter- and intra-class variability.** In medical imaging, samples within the same class (e.g., benign lesions, specific cell types) often appear highly homogeneous, especially after standard preprocessing. While this reduces within-class spread, it also compresses class prototypes into overly compact regions of the feature space. As new classes are introduced, these compact prototypes become highly correlated, increasing the risk of representation collapse where different classes cannot be reliably distinguished.

**(2) High inter-domain shifts.** Medical data are acquired across diverse scanners, protocols, and institutions, leading to substantial distribution shifts in texture, resolution, and intensity. Such variability fragments class distributions and destabilizes the alignment of prototypes, resulting in domain misalignment. The same class may occupy multiple disjoint regions in feature space, undermining the assumption of stable class anchors that many CIL methods rely on.

**Why it matters.** In natural image benchmarks such as CIFAR100 or ImageNet, intra-class variability is typically high: a *dog* class can include many breeds, poses, and backgrounds, which encourages feature learning that captures broad semantic diversity. Domain shifts also exist (e.g., between natural photos, sketches in ImageNet-R, or adversarial corruptions in ImageNet-A), but these shifts tend to be more semantic or stylistic in nature rather than stemming from systematic acquisition differences. In contrast, medical imaging presents the opposite challenge. Intra-class variability is often low, as samples from the same condition (e.g., red blood cells, COVID-positive CT scans) appear visually homogeneous, compressing class distributions in feature space. At the same time, domain shifts are severe and systematic, caused by differences in scanners, acquisition protocols, and institutions, which fragment same-class features into disjoint regions. This interplay of compact intra-class clusters and strong inter-domain scatter magnifies representation collapse and domain misalignment, explaining why methods effective in natural images may falter in medical CIL.

## B    PARAMETER-EFFICIENT TUNING (PET) METHODS

We briefly summarize the formulations of three PET methods commonly used in existing PTM-based CIL methods:

**Visual Prompt Tuning (VPT)**    (Jia et al., 2022). VPT introduces a set of learnable prompts $\mathbf{P} \in \mathbb{R}^{p \times d_0}$ prepended to the input embeddings $\mathbf{x}_e$, forming the extended input $[\mathbf{P}, \mathbf{x}_e]$. The pre-trained ViT backbone is frozen, and only the prompts and classification head are updated:

$$\Theta = \theta_{\mathbf{P}} \cup \theta_W. \tag{5}$$

**Scale and Shift (SSF)**    (Lian et al., 2022). SSF applies learnable scaling and shifting to the input of each transformer block. For each input $\mathbf{x}_i \in \mathbb{R}^{L \times d_0}$, the output is computed as:

$$\mathbf{x}_o = \gamma \odot \mathbf{x}_i + \beta, \tag{6}$$

where $\gamma, \beta \in \mathbb{R}^{d_0}$ and $\odot$ is the Hadamard product. Only the SSF parameters and classifier are optimized:

$$\Theta = \theta_{\text{SSF}} \cup \theta_W. \tag{7}$$

**ViT-Adapter**    (Chen et al., 2022). Adapter inserts a bottleneck module into the MLP blocks of ViT. The module consists of a down-projection $W_{\text{down}} \in \mathbb{R}^{k \times \hat{k}}$, a ReLU activation function $\sigma$, and an up-projection $W_{up} \in \mathbb{R}^{\hat{k} \times k}$. Following (Tan et al., 2024; Zhou et al., 2024b), we augment the ViT's multilayer perceptron (MLP) layers with adapters with projected dimension $\hat{k}$ equal to 64 (Chen et al., 2022). Let $x_{in}$ be the input of the MLP layer, the output of the adapter-equipped MLP is:

$$x_{out} = MLP(x_{in}) + \sigma \left( x_{in} * W_{\text{down}} \right) * W_{up}, \tag{8}$$

where $*$ denotes matrix multiplication. During training, the ViT backbone remains frozen, and the optimizable parameters can be denoted as: $\Theta = \theta_{W_{\text{down}}} \cup \theta_{W_{\text{up}}} \cup \theta_W$.

## C  THREE TYPES OF CLASSIFIERS

This section provides detailed formulations for three types of classifiers.

**Fully-connected (FC) classifier.**   First, given the extracted feature $\phi(x)$, the softmax-based logits for the standard FC classifier can be formulated as:

$$z = W^\top \phi(\mathbf{x}), \tag{9}$$

where $W \in \mathbb{R}^{d_0 \times N}$ represents the trainable weight matrix for $N$ classes learnt so far, and $z \in \mathbb{R}^N$ contains the unnormalized scores for each class. These logits are typically passed through a softmax function to obtain class probabilities: $p_y = \frac{\exp(z_y)}{\sum_{n=1}^{N} \exp(z_n)}$, where $p_y$ denotes the predicted probability for class $y$.

**Class prototype (CP)-based classifier.**   The CP-based classifier (Zhou et al., 2024a) computes class prototypes as:

$$p_i = \frac{1}{K} \sum_{j=1}^{|D^b|} \mathbb{I}(y_j = i)\phi(x_j), \tag{10}$$

where $K$ denotes the number of samples belonging to class $i$ within the current incremental learning batch $D^b$ and $\mathbb{I}(y_j = i)$ is an indicator function that equals 1 if the sample $x_j$ belongs to class $i$, and 0 otherwise. Following the nearest class mean (NCM) principle (Mensink et al., 2013), classifier assigns new samples to the class whose prototype is closest in the feature space during inference and measures the cosine similarity between the sample embedding and class prototype:

$$z = \frac{\phi(x)^\top p_i}{\|\phi(x)\|\|p_i\|}, i = 1, \dots, N, \tag{11}$$

where $N$ is the total number of classes learned up to the current session.

**Analytical decorrelated prototype-based classifier.**   Analytical learning refers to approaches where classifier parameters are computed in closed form, rather than updated through iterative gradient descent. A common formulation is ridge regression, which yields a stable closed-form solution with built-in regularization. This analytic view is attractive in continual learning because it avoids repeated re-optimization, reduces computational overhead, and provides stability across tasks. In our case, the analytical decorrelated prototype-based classifier can be derived as a ridge regression solution over the projected features $h(x) \in \mathbb{R}^{d_1}$. Specifically, we first compute the feature autocorrelation (Gram) matrix:

$$G = \sum_{t} \sum_{n=1}^{N_t} h_{t,n} h_{t,n}^\top \in \mathbb{R}^{d_1 \times d_1}. \tag{12}$$

We then compute another matrix $C_p$ that consists of the accumulated sum of projected features corresponding to the same class labels, defined by

$$C_p = \sum_{t} \sum_{n=1}^{N_t} h_{t,n} y_{t,n}^\top \in \mathbb{R}^{d_1 \times |Y_t|}. \tag{13}$$

The predicted logits are then computed as:

$$z = h W_{adc} = h(G + \beta I)^{-1} C_p \in \mathbb{R}^{|Y_t|}, \tag{14}$$

with $\beta$ the ridge regularization parameter selected by cross-validation-based optimization, which helps stabilize matrix inversion and prevent overfitting over continual learning, and $I$ the identity matrix. As such, $W_{adc}$ can be interpreted as decorrelated class prototypes.

In this formulation, $W_{adc}$ can be directly interpreted as a set of class prototypes. The "decorrelation" property (see Appendix F) emerges naturally from the ridge regression solution, since the inversion of $(G + \beta I)$ whitens correlated features. Thus, without introducing extra mechanisms, the classifier yields decorrelated prototypes that preserve discriminability across tasks and mitigate representation collapse in continual learning.

## D  CLOSED-FORM SOLUTION DERIVATION OF ANALYTICAL DECORRELATED CLASSIFIER

To derive the analytical classifier weights, we start from the ridge regression objective in Eq. 1:

$$\arg \min_{W_{adc}} \|Y - HW_{adc}\|_F^2 + \beta \|W_{adc}\|_F^2 , \tag{15}$$

where $H \in \mathbb{R}^{N \times d_1}$ is the stacked matrix of all projected sample features; $Y \in \mathbb{R}^{N \times |Y_t|}$ contains the corresponding one-hot labels; $N = \sum_{t=1}^{T} N_t$ is the total number of samples accumulated up to the current task; $\|\cdot\|_F$ is the Frobenius norm; and $\beta$ is the ridge regularization parameter selected by cross-validation-based optimization. Taking the gradient of this objective with respect to $W_{adc}$ gives:

$$\frac{\partial}{\partial W_{adc}}\left(\|Y - HW_{adc}\|_F^2 + \beta\|W_{adc}\|_F^2\right) = -2H^\top(Y - HW_{adc}) + 2\beta W_{adc}. \tag{16}$$

Setting this gradient to zero yields:

$$-2H^\top Y + 2H^\top HW_{adc} + 2\beta W_{adc} = 0, \tag{17}$$

which can be rearranged into:

$$\left(H^\top H + \beta I\right)W_{adc} = H^\top Y. \tag{18}$$

Solving for $W_{adc}$ gives the closed-form solution:

$$\hat{W}_{adc} = \left(H^\top H + \beta I\right)^{-1} H^\top Y. \tag{19}$$

Finally, expressing this solution in terms of the feature autocorrelation (Gram) matrix $G = H^\top H$ and the class-accumulated feature matrix $C_p = H^\top Y$ yields:

$$\hat{W}_{adc} = (G + \beta I)^{-1} C_p \in \mathbb{R}^{d_1 \times |Y_t|}, \tag{20}$$

which is exactly Eq. 2.

## E  PRIVACY-PRESERVING NATURE OF ANALYTICAL DECORRELATED CLASSIFIER

A potential concern is whether maintaining the accumulated statistics $G$ and $C_p$ may compromise data privacy, since they are derived from individual features $h(\mathbf{x})$. We emphasize that both matrices are statistical summaries rather than raw features: (i) The feature autocorrelation (Gram) matrix $G$ only encodes second-order relationships (feature covariance structure) without storing individual feature vectors. (ii) The class-accumulated feature matrix $C_p$ aggregates features at the class level, equivalent to storing class prototypes rather than per-sample embeddings. As such, the stored information reflects only global distributional properties of the feature space and does not allow direct reconstruction of individual patient images or embeddings. In high-sample regimes ($N \gg d_1$), the contribution of any single sample is negligible, making reverse-engineering practically infeasible. This property is consistent with prior work in privacy-preserving machine learning and federated learning, where sharing first- or second-order statistics has been considered substantially safer than transmitting raw data. Compared with conventional rehearsal-based methods that store raw features or exemplars, our approach is therefore inherently more privacy-preserving, since it requires only compact statistical summaries rather than patient-level data.

## F  PROOF OF THE DECORRELATION PROPERTY

**Analytical Formulation.** The analytical decorrelated classifier (ADC) is defined as:

$$\hat{W}_{adc} = (G + \beta I)^{-1} C_p, \tag{21}$$

where $G = \sum_n h_n h_n^\top$ is the feature autocorrelation (Gram) matrix, $C_p = \sum_n h_n y_n^\top$ is the class-accumulated feature matrix, and $\beta$ is the ridge regularization parameter.

**Spectral Reweighting.** Let $G = U\Lambda U^\top$ be the eigendecomposition of $G$, with $\Lambda = \mathrm{diag}(\lambda_1, \ldots, \lambda_{d_1})$. Then:

$$(G + \beta I)^{-1} = U \, \mathrm{diag}\left( \tfrac{1}{\lambda_1 + \beta}, \ldots, \tfrac{1}{\lambda_{d_1} + \beta} \right) U^\top. \tag{22}$$

This expression shows that each eigendirection of the feature space is rescaled by an inverse factor of its variance.

- Directions with large variance (often corresponding to domain-shared nuisance factors) are suppressed.
- Directions with small variance (often containing class-specific information) are relatively amplified.

This behavior is closely related to whitening, which applies $G^{-1/2}$ to normalize feature covariance to the identity. While $(G + \beta I)^{-1}$ is not an exact whitening operator, it achieves a whitening-like effect: balancing the contributions of eigendirections and reducing correlations across prototypes. The ridge term $\beta$ further prevents instability when $G$ is ill-conditioned, which is common under scarce and heterogeneous medical imaging data.

**Prototype Decorrelation.** Let $w_i$ and $w_j$ denote two columns of $\hat{W}_{adc}$ (the prototypes for classes $i$ and $j$). Their covariance is:

$$\mathrm{Cov}(w_i, w_j) = C_{p,i}^\top U (\Lambda + \beta I)^{-2} U^\top C_{p,j}. \tag{23}$$

Since $(\Lambda + \beta I)^{-2}$ down-weights directions with large eigenvalues, the dominant shared components that typically drive high inter-class correlation are strongly suppressed. As a result, the learned prototypes are decorrelated and more discriminative across tasks.

In summary, the ADC achieves decorrelation by:

- reweighting feature eigendirections in a whitening-like manner,
- suppressing shared nuisance components while emphasizing discriminative cues, and
- stabilizing inversion with ridge regularization under limited data.

This explains theoretically why ADC prototypes are less correlated, more robust to domain shifts, and consistently outperform the standard prototype-based classifier (Table 5).

Table 5: Preliminary study on analytical decorrelated classifier across representative datasets. PTM: ViT-B/16-IN21K.

| Dataset | $f_{\theta_{\mathrm{cls}}} \to f_{\theta_{\mathrm{cls}}}$ (D) |
|---|---|
| IN-A | 48.39 → 54.18 |
| CIFAR | 81.28 → 86.09 |
| MedMNIST-Sub | 50.45 → 66.44 |

# G ADDITIONAL EXPERIMENTS

## G.1 RESULTS ON DIFFERENT RANDOM MATRIX

Table 6: Experimental results on different initialization methods for random matrix under CIL settings across general-domain and medical datasets ($d_1=5d_0$). "K." stands for Kaiming initialization, "X" stands for Xaiver initialization, "U." stands for uniform distribution, "N." stands for normal distribution, and the number is the parameter for initialization. Note that all experiments are conducted without first-stage adaptation (Original ViT w/ B&LRP), except for the full RA-LDL.

| Methods | CIFAR100 | | IN-A | | CUB | | IN-R | |
|---|---|---|---|---|---|---|---|---|
| | $Acc_{Avg}$ (%) | $Acc_{Last}$ (%) | $Acc_{Avg}$ (%) | $Acc_{Last}$ (%) | $Acc_{Avg}$ (%) | $Acc_{Last}$ (%) | $Acc_{Avg}$ (%) | $Acc_{Last}$ (%) |
| Standard Gaussian (Full RA-LDL) | **94.93** | **91.55** | **70.84** | **60.51** | **93.03** | **89.14** | **83.64** | **78.17** |
| Standard Gaussian | 91.66 | 87.61 | 64.78 | 53.52 | 92.63 | 89.07 | 75.94 | 69.94 |
| K. U. $\sqrt{5}$ | 91.95 | 87.73 | 62.97 | 54.44 | 91.40 | 86.56 | 74.93 | 69.52 |
| K. U. 0 | 91.90 | 87.61 | 62.76 | 54.42 | 92.50 | 88.51 | 74.63 | 69.10 |
| K. N. 0 | 91.82 | 87.69 | 63.44 | 53.39 | 92.69 | 88.82 | 74.73 | 69.30 |
| X. U. 0 | 91.87 | 87.58 | 63.73 | 55.37 | 92.44 | 88.35 | 74.94 | 69.98 |
| X. N. 0 | 91.81 | 87.66 | 63.67 | 53.85 | 92.63 | 88.68 | 74.60 | 69.75 |
| Methods | MedMNIST-Sub | | Skin8 | | COVID | | Blood | |
| | $Acc_{Avg}$ (%) | $Acc_{Last}$ (%) | $Acc_{Avg}$ (%) | $Acc_{Last}$ (%) | $Acc_{Avg}$ (%) | $Acc_{Last}$ (%) | $Acc_{Avg}$ (%) | $Acc_{Last}$ (%) |
| Standard Gaussian (Full RA-LDL) | **84.79** | **70.60** | **75.80** | 63.83 | 95.98 | 88.04 | **98.07** | **97.76** |
| Standard Gaussian | 84.30 | 70.29 | 75.52 | 61.03 | **95.47** | **90.30** | 98.06 | 97.74 |
| K. U. $\sqrt{5}$ | 68.78 | 58.15 | 75.29 | **63.96** | 85.77 | 70.94 | 45.85 | 19.31 |
| K. U. 0 | 81.64 | 67.09 | 75.15 | 61.99 | 92.31 | 79.17 | 86.22 | 83.71 |
| K. N. 0 | 82.12 | 68.06 | 74.92 | 62.88 | 94.12 | 81.94 | 92.87 | 91.71 |
| X. U. 0 | 81.36 | 66.79 | 75.12 | 61.56 | 91.77 | 81.73 | 86.36 | 82.18 |
| X. N. 0 | 81.68 | 67.41 | 75.76 | 62.55 | 95.81 | 89.60 | 92.95 | 91.47 |

## G.2 RESULTS ON DIFFERENT PROJECTION DIMENSIONS

Table 7: Preliminary analysis results on the random projection (RP) dimension $d_1$: The average and last performance on eight CIL datasets from both general and medical fields using **ViT-B/16-IN21K** as the backbone. All experiments are conducted using the same seed for a fair comparison. The best performance is highlighted in **bold**.

| Dataset | First-stage Adaptation | $Acc_{Last}$ | | | |
|---|---|---|---|---|---|
| | | $d_1=d_0$ | $d_1=5d_0$ | $d_1=10d_0$ | $d_1=15d_0$ |
| CIFAR | ✓ | 90.15 | 91.55 | 91.97 | **92.17** |
| | ✗ | 85.61 | 87.63 | 88.60 | **89.11** |
| IN-A | ✓ | 55.89 | 60.51 | **61.09** | 60.83 |
| | ✗ | 49.11 | 53.52 | 57.08 | **57.27** |
| CUB | ✓ | 85.71 | 89.14 | **89.82** | 89.78 |
| | ✗ | 85.33 | 89.07 | 89.82 | **90.12** |
| OB | ✓ | 73.47 | 78.05 | 78.76 | **80.02** |
| | ✗ | 71.78 | 75.91 | 77.63 | **78.50** |
| Skin8 | ✓ | 57.45 | 62.49 | **65.96** | 63.69 |
| | ✗ | 56.74 | 60.85 | 62.98 | **66.24** |
| MedMNIST-Sub | ✓ | 67.70 | 70.60 | 72.14 | **72.55** |
| | ✗ | 66.22 | 70.29 | 71.40 | **71.43** |
| COVID (CT&X-rays) | ✓ | 83.68 | 88.04 | 88.45 | **88.99** |
| | ✗ | 84.42 | 90.30 | **90.32** | 87.41 |
| Blood | ✓ | 96.34 | **97.76** | 97.70 | 97.74 |
| | ✗ | 95.99 | **97.74** | 97.62 | 97.58 |

## G.3 SENSITIVITY EVALUATION ON RANK $r$

Table 8: Sensitivity evaluation on rank $r$ in Original ViT w/ B&LRP.

| $r$ | CUB | | IN-A | | Skin8 | | MedMNIST-Sub | |
|---|---|---|---|---|---|---|---|---|
| | $Acc_{Avg}$ (%) | $Acc_{Last}$ (%) | $Acc_{Avg}$ (%) | $Acc_{Last}$ (%) | $Acc_{Avg}$ (%) | $Acc_{Last}$ (%) | $Acc_{Avg}$ (%) | $Acc_{Last}$ (%) |
| $r = 64$ | 92.63 | 89.07 | 64.78 | 53.52 | 75.52 | 60.85 | 84.30 | 70.29 |
| $r = 128$ | 92.72 | 89.12 | 65.33 | 55.37 | 75.37 | 60.86 | 84.34 | 70.30 |
| $r = 256$ | 92.69 | 89.12 | 64.90 | 55.43 | 75.42 | 60.99 | 84.32 | 70.31 |

## G.4 EXPERIMENTS COMPARED TO RECENT METHODS IN GENERAL-DOMAIN CIL

Table 9: Experimental results (10 tasks) compared to recent PTM-based CIL methods on general-domain benchmarks. "†" denotes reliance on replayed data. The best and second-best results are marked in **red** and *blue*, respectively.

| Methods | General Domain | | | | | | | |
| --- | --- | --- | --- | --- | --- | --- | --- | --- |
| | CIFAR100 | | IN-A | | CUB | | IN-R | |
| | $Acc_{Avg}$ (%) | $Acc_{Last}$ (%) | $Acc_{Avg}$ (%) | $Acc_{Last}$ (%) | $Acc_{Avg}$ (%) | $Acc_{Last}$ (%) | $Acc_{Avg}$ (%) | $Acc_{Last}$ (%) |
| Joint-training | - | 93.06 | - | 57.60 | - | 88.20 | - | 81.43 |
| Finetune | 39.12 | 19.85 | 20.03 | 9.82 | 26.89 | 14.25 | 22.54 | 11.60 |
| L2P (Wang et al., 2022c) | 90.94 | 85.79 | 52.49 | 44.17 | 79.63 | 68.19 | 77.98 | 72.47 |
| DualPrompt (Wang et al., 2022b) | 89.59 | 85.31 | 57.63 | 46.28 | 79.80 | 69.04 | 75.05 | 69.10 |
| CodaPrompt (Smith et al., 2023) | 91.73 | 87.70 | 58.22 | 45.95 | 80.53 | 72.69 | 77.99 | 73.06 |
| LAE (Gao et al., 2023) | 85.08 | 79.36 | 42.48 | 30.09 | 73.87 | 61.03 | 68.82 | 61.28 |
| SLCA (Zhang et al., 2023a) | 93.32 | 91.80 | 69.10 | *61.23* | 90.92 | 85.54 | 81.82 | 77.95 |
| SSIAT (Tan et al., 2024) | 94.71 | **92.20** | *70.86* | **62.54** | 91.37 | **89.78** | **84.30** | **79.82** |
| EASE (Zhou et al., 2024b) | 91.51 | 85.80 | 65.34 | 55.04 | 88.08 | 81.38 | 82.60 | 77.35 |
| MOS (Sun et al., 2025) | 93.84 | 90.88 | 68.28 | 58.92 | **93.11** | *89.22* | 82.81 | 77.82 |
| SimpleCIL (Zhou et al., 2024a) | 91.51 | 85.80 | 59.67 | 52.60 | 91.85 | 86.77 | 61.18 | 54.35 |
| ADAM-Adapter (Zhou et al., 2024a) | 90.93 | 85.82 | 65.13 | 54.31 | 92.14 | 87.53 | 75.12 | 66.72 |
| Original ViT w/ B | 91.91 | 87.81 | 64.68 | 53.25 | 92.60 | 89.01 | 75.90 | 69.92 |
| Adapted ViT w/ B | **95.37** | *92.13* | **71.12** | 60.50 | 93.00 | 89.03 | 83.33 | 77.80 |
| Original ViT w/ B&LRP (**RA-LDL**) | 91.88 | 87.63 | 64.78 | 53.52 | 92.63 | 89.07 | 75.94 | 69.94 |
| Adapted ViT w/ B&LRP (**RA-LDL**) | *94.93* | 91.55 | 70.84 | 60.51 | *93.03* | 89.14 | *83.64* | *78.17* |

## G.5 EXPERIMENTS ON CROSS-DATASET LONG-TERM CONTINUAL LEARNING BENCHMARK

Table 10: Experimental results on cross-dataset long-term continual learning benchmark (order: Skin8 → COVID → MedMNIST → BLOOD), following the original intra-dataset task splits in Table 2.

| Framework | $Acc_{Avg}$ | $Acc_{Last}$ |
| --- | --- | --- |
| Joint Training (upper bound) | – | 71.65 |
| SimpleCIL | 57.50 | 55.41 |
| ADAM-Adapter | 63.18 | 55.52 |
| SSIAT | 53.97 | 28.41 |
| MOS | 70.66 | 62.05 |
| **RA-LDL (ours)** | **74.56** | **71.52** |

## G.6 SENSITIVITY EVALUATION ON TASK NUMBER

Table 11: Results with varying task numbers on MedMNIST-Sub dataset.

| Method | Task Number | $Acc_{Avg}$ (%) | $Acc_{Last}$ (%) |
| --- | --- | --- | --- |
| SimpleCIL (Zhou et al., 2024a) | 4 | 68.07 | 50.63 |
| SimpleCIL (Zhou et al., 2024a) | 8 | 51.41 | 50.44 |
| MOS (Sun et al., 2025) | 4 | 74.59 | 51.80 |
| MOS (Sun et al., 2025) | 8 | 80.96 | 61.83 |
| RA-LDL | 4 | 84.79 | 70.60 |
| RA-LDL | 8 | **86.71** | **70.80** |

# H PROOF OF RANDOM ANCHORS FOR FEATURE CHARACTERISTICS PRESERVATION AND ENHANCEMENT

We formally prove that random projections guided by the Johnson–Lindenstrauss lemma effectively preserve original feature characteristics and, when used to increase dimensionality, can improve the structural and statistical properties of features extracted from pre-trained models.

## H.1 JOHNSON-LINDENSTRAUSS LEMMA

**Lemma 1** (Johnson–Lindenstrauss Lemma). *Given a distortion tolerance $0 < \epsilon < 1$ and a finite set of points $x_1, x_2, \ldots, x_m \subseteq \mathbb{R}^{d_0}$, there exists a random linear projection $B \in \mathbb{R}^{d_0 \times d_1}$, where the target dimension $d_1$ satisfies*

$$d_1 = O\left(\frac{\log m}{\epsilon^2}\right), \tag{24}$$

*such that for all $i, j \in \{1, \ldots, m\}$, the following inequality holds with high probability:*

$$(1 - \epsilon)\|x_i - x_j\|^2 \leq \|B^\top(x_i - x_j)\|^2 \leq (1 + \epsilon)\|x_i - x_j\|^2. \tag{25}$$

The lemma demonstrates that random linear projection approximately preserves pairwise distances with high probability. As proved in the following subsections (H.3), pairwise distances between all data points are uniformly preserved with high probability, maintaining the original geometric relationships and avoiding feature confusion. Similarly, the expected covariance and inner products scale linearly, and cosine similarities are preserved in expectation.

Next, we discuss the benefits of increasing feature dimensionality through random projections.

## H.2 Benefits of Random Projection into Higher Dimensions

**Theorem 2** (Benefits of Random Projection into Higher Dimensions). *Consider a random linear projection defined by a Gaussian matrix $B \in \mathbb{R}^{d_0 \times d_1}$, with $d_1 > d_0$ and entries independently sampled as:*

$$B_{ij} \sim \mathcal{N}\left(0, \frac{1}{d_1}\right) \tag{26}$$

*Given our extracted feature $\phi(\mathbf{x}) \in \mathbb{R}^{d_0}$, the projection $h_{RA}(\mathbf{x}) = \mathbf{B}^\top \phi(\mathbf{x})$ into higher dimension $d_1$ improves isotropy of feature distributions and reduces intra-class variance relative to inter-class variance, enhancing linear separability.*

*Proof.* We analyze the projected features $h_{RA}(\mathbf{x}) = \mathbf{B}^\top \phi(\mathbf{x}) \in \mathbb{R}^{d_1}$.

First, we consider the inner product of two distinct vectors $h_{RA}(\mathbf{x_i}), h_{RA}(\mathbf{x_j})$ after projection:

$$\mathbb{E}\left[h_{RA}(\mathbf{x_i})^\top h_{RA}(\mathbf{x_j})\right] = h_{RA}(\mathbf{x_i})^\top \mathbb{E}\left[BB^\top\right] h_{RA}(\mathbf{x_j}) = 0, \tag{27}$$

due to independence and zero-mean properties of the random Gaussian projection matrix $B_{ij}$. Thus, the random projection helps decorrelate features and increase isotropy. Next, we examine the covariance structure within each class. Suppose the intra-class covariance matrix of the original features $\phi(x)$ is denoted as $\Sigma \in \mathbb{R}^{d_0 \times d_0}$. After the projection, the covariance matrix becomes:

$$\mathbb{E}\left[B^\top \Sigma B\right] = \frac{\mathrm{Tr}(\Sigma)}{d_1} I_{d_1}. \tag{28}$$

This indicates that projecting to a higher dimension $d_1$ makes intra-class distributions more isotropic and reduces per-dimension variance. Simultaneously, the expected squared distance between class means $\mu_1, \mu_2$ is preserved:

$$\mathbb{E}\left[\left\|B^\top(\mu_1 - \mu_2)\right\|^2\right] = (\mu_1 - \mu_2)^\top \mathbb{E}\left[BB^\top\right](\mu_1 - \mu_2) = \|\mu_1 - \mu_2\|^2. \tag{29}$$

Thus, increasing the dimension from $d_0$ to $d_1$ effectively reduces intra-class variance without affecting the inter-class separation, thereby improving class separability. □

## H.3 Proof of Random Anchor Preserving Feature Characteristics

**i) Pairwise Distances.**

*Proof.* Consider a random Gaussian projection matrix $B \in \mathbb{R}^{d_0 \times d_1}$, with each element independently drawn from a Gaussian distribution::

$$B_{ij} \sim \mathcal{N}\left(0, \frac{1}{d_1}\right), \quad \mathbb{E}[B_{ij}] = 0, \quad \mathrm{Var}(B_{ij}) = \frac{1}{d_1}. \tag{30}$$

Consider two feature vectors $\phi(x_i), \phi(x_j) \in \mathbb{R}^{d_0}$, and define their difference as: $v = \phi(x_i) - \phi(x_j)$. Then, the squared norm of the projected difference is given by:

$$\|B^\top v\|^2 = \sum_{p=1}^{d_1}\left(\sum_{q=1}^{d_0} B_{p,q}\, v_q\right)^2, \tag{31}$$

Define the projected component for each $p$ as:

$$Y_p = \sum_{q=1}^{d_0} B_{p,q} v_q. \tag{32}$$

Since the entries of $B$ are zero-mean and independent, we have:

$$\mathbb{E}[Y_p] = 0, \quad \mathrm{Var}(Y_p) = \mathbb{E}\left[\left(\sum_{q=1}^{d_0} B_{p,q} v_q\right)^2\right] = \frac{\|v\|^2}{d_1}. \tag{33}$$

Applying concentration inequalities (such as Gaussian concentration), we have, simultaneously for all pairs $(i, j)$, with probability at least $1 - \delta$:

$$(1 - \varepsilon)\|v\|^2 \leq \|B^\top v\|^2 \leq (1 + \varepsilon)\|v\|^2, \tag{34}$$

provided the projection dimension satisfies:

$$d_1 \geq C\,\epsilon^{-2}\,\log\left(\frac{m^2}{\delta}\right) = O\left(\frac{\log m}{\epsilon^2}\right), \tag{35}$$

where $C$ is a suitable constant and $m$ is the number of data points. This shows that pairwise distances are approximately preserved. $\qquad\square$

### ii) Variance and Covariance Preservation (Isserlis formula).

*Proof.* Given the random projection matrix $B$ defined above, the expected covariance and inner products scale linearly, and cosine similarities are exactly preserved in expectation. Let $\Sigma \in \mathbb{R}^{d_0 \times d_0}$ be the covariance matrix of the input features. The trace of the projected covariance matrix is:

$$\mathbb{E}\big[\mathrm{tr}(B^\top \Sigma B)\big] = \sum_{p=1}^{d_1}\sum_{q=1}^{d_0}\sum_{r=1}^{d_0} \Sigma_{q,r}\,\mathbb{E}[B_{q,p}B_{r,p}]. \tag{36}$$

By independence and zero-mean, only the diagonal terms $q = r$ contribute, giving:

$$\mathbb{E}[B_{q,p}B_{r,p}] = \begin{cases} 1/d_1, & \text{if } q = r, \\ 0, & \text{otherwise,} \end{cases} \tag{37}$$

so the expectation simplifies to:

$$\mathbb{E}\big[\mathrm{tr}(B^\top \Sigma B)\big] = \frac{d_1}{d_0}\,\mathrm{tr}(\Sigma). \tag{38}$$

$\qquad\square$

### iii) Inner Products and Norms (Isserlis formula).

*Proof.* Consider two vectors $u, v \in \mathbb{R}^{d_0}$ and their projected inner product:

$$\langle Bu, Bv\rangle = \sum_{p=1}^{d_1}(Bu)_p(Bv)_p = \sum_{p=1}^{d_1}\sum_{i=1}^{d_0}\sum_{j=1}^{d_0} B_{i,p}u_i\,B_{j,p}v_j. \tag{39}$$

Taking expectation:

$$\mathbb{E}\big[\langle Bu, Bv\rangle\big] = \sum_{p=1}^{d_1}\sum_{i=1}^{d_0}\sum_{j=1}^{d_0} u_i v_j\,\mathbb{E}\big[B_{i,p}B_{j,p}\big]. \tag{40}$$

Using the same independence argument as above:

$$\mathbb{E}\big[\langle Bu, Bv\rangle\big] = \frac{d_1}{d_0}\,\langle u, v\rangle. \tag{41}$$

Similarly, we also verify norm preservation by setting $u = v$:

$$\mathbb{E}\big[\|Bu\|^2\big] = \mathbb{E}\big[\langle Bu, Bu\rangle\big] = \frac{d_1}{d_0}\|u\|^2. \tag{42}$$

### iv) Cosine Similarity.
Since both the inner product and squared norms scale by $\frac{d_1}{d_0}$, the cosine similarity remains unchanged:

$$\mathbb{E}[\cos\theta'] = \frac{\mathbb{E}[\langle Bu, Bv\rangle]}{\sqrt{\mathbb{E}[\|Bu\|^2]\,\mathbb{E}[\|Bv\|^2]}} = \frac{\frac{d_1}{d_0}\langle u, v\rangle}{\sqrt{\frac{d_1}{d_0}\|u\|^2\,\frac{d_1}{d_0}\|v\|^2}} = \cos\theta. \tag{43}$$

Thus, the random linear projection preserves covariance, inner product structures, and angular relationships in expectation. $\qquad\square$

# I    PROOF OF LRP BENEFITS

## I.1    OPTIMALITY OF RESIDUAL LRP COMPLEMENT TO RA

**Theorem 3** (Optimality of Residual LRP Complement to RA). *Let $\phi(\mathbf{x}) \in \mathbb{R}^{d_0}$ be the feature from the PTM, and let $h_{RA}(\mathbf{x}) = \mathbf{B}^\top \phi(\mathbf{x})$, with $\mathbf{B} \in \mathbb{R}^{d_0 \times d_1}$ frozen. Suppose the downstream domain exhibits additional residual variation $\Sigma_\Delta$ such that the intra-class covariance becomes $\Sigma_{target} = \Sigma + \Sigma_\Delta$. Then a trainable low-rank correction $h_{LRP}(\mathbf{x})$, trained in the first session, reduces intra-domain variance in the RA-projected space while preserving inter-class distances.*

*Proof.* After RA projection, the intra-class covariance under domain shift is:

$$\mathbb{E}[\mathbf{B}^\top \Sigma_{target} \mathbf{B}] = \frac{\text{Tr}(\Sigma + \Sigma_\Delta)}{d_1} I_{d_1}. \tag{44}$$

Training $h_{LRP}$ optimizes:

$$\min_{\mathbf{W}_1, \mathbf{W}_2} \mathbb{E}\left[\|h_{LRP}(\mathbf{x}) - \Delta(\mathbf{x})\|^2\right], \tag{45}$$

where $\Delta(\mathbf{x})$ denotes the domain shift correction. Due to the low-rank constraint, the learned correction emphasizes dominant residual variations. Thus, the residual term complements RA by explicitly correcting domain-induced distortions without altering the preserved global geometry, reducing intra-class spread while maintaining inter-class structure. $\square$

## I.2    RESIDUAL IMPROVES CLASSIFICATION MARGIN

**Theorem 4** (Residual Improves Classification Margin). *Let $\mu_1, \mu_2 \in \mathbb{R}^{d_0}$ be class means in the feature space. Then the expected squared margin after adding a first-session-trained LRP residual satisfies:*

$$\mathbb{E}\left[\|h(\mu_1) - h(\mu_2)\|^2\right] > \mathbb{E}\left[\|h_{RA}(\mu_1) - h_{RA}(\mu_2)\|^2\right], \tag{46}$$

*assuming the residual approximates domain-induced mean shift $\Delta\mu_{domain} \neq 0$.*

*Proof.* Let:

$$h(\mu) = \mathbf{B}^\top \mu + h_{LRP}(\mu). \tag{47}$$

Then:

$$\begin{aligned}
\mathbb{E}[\|h(\mu_1) - h(\mu_2)\|^2] &= \mathbb{E}[\|\mathbf{B}^\top(\mu_1 - \mu_2) + h_{LRP}(\mu_1) - h_{LRP}(\mu_2)\|^2] \\
&= \|\mu_1 - \mu_2\|^2 + \|h_{LRP}(\mu_1) - h_{LRP}(\mu_2)\|^2 \\
&\quad + 2\langle \mathbf{B}^\top(\mu_1 - \mu_2), h_{LRP}(\mu_1) - h_{LRP}(\mu_2)\rangle.
\end{aligned} \tag{48}$$

The cross-term expectation is zero due to the independence of $\mathbf{B}$ and $h_{LRP}$. If $h_{LRP}$ approximates the domain-specific mean shift, the added term is nonzero, and the margin increases. $\square$

# J    LIMITATIONS AND SOCIETAL IMPACTS

(i) Scope: Our evaluation focuses on 2D classification; extending to 3D, multi-modal, and segmentation tasks remains future work. (ii) Novelty: While certain components have been explored in general-domain studies, the key contribution of this work lies in demonstrating that a simple yet carefully integrated strategy can effectively address medical-specific challenges—and even outperform more complex designs in medical class-incremental learning. (iii) Robustness: While generally stable, RA-LDL shows order sensitivity on heterogeneous datasets. Further calibration could address this. (iv) Backbone Dependence: Performance is bounded by PTM quality; better PTMs should further enhance results. (v) Impact: The lightweight design lowers computational cost and facilitates privacy-friendly deployment, but careful clinical validation is essential.

In summary, RA-LDL is a foundational step: not a final solution, but evidence that minimalist recalibration can be surprisingly effective in medical continual learning.

## K LLM USAGE

The authors utilized LLMs as a writing assistant during the preparation of this manuscript. The role of it was primarily focused on tasks related to language refinement and improving the clarity of the narrative. Its application included: (1) enhancing the conciseness and academic tone of sentences and paragraphs; (2) restructuring complex sentences for better readability; and (3) brainstorming alternative phrasings for key scientific arguments.

It is important to emphasize that all core scientific ideas, experimental design, results, and conclusions were exclusively developed by the human authors. The LLM served as a sophisticated editing and brainstorming tool. All text generated or modified by the LLM was critically reviewed, edited, and ultimately approved by the authors to ensure it accurately reflected our original intent and scientific findings.

