# OpenReview forum: "Random Anchors with Low-rank Decorrelated Learning: A Minimalist Pipeline for Class-Incremental Medical Image Classification"
_ICLR.cc/2026/Conference — ICLR 2026 Poster_

### Official Review · Reviewer_zaiN · 2025-10-29

**Soundness:** 3
**Presentation:** 2
**Contribution:** 2
**Rating:** 4
**Confidence:** 4

**Summary:**

In this paper, the authors propose a Random Anchors with Low-rank Decorrelated Learning (RA-LDL) approach to address Class-incremental Learning (CIL) in medical image diagnosis. The authors validate their idea with two base models (ViT-B/16-IN21K and BiomedCLIP) and four datasets (Covid, Blood, Skin8, MedMNIST-sub). The gains of RA-LDL are clear.

**Strengths:**

1. The experiments are comprehensive: the authors include various types of medical images and popular base models. In the comparisons, the authors also include most milestone methods in class incremental learning. All these experiments show a clear gain for RA-LDL.
2. Analytical closed-form decorrelated learning sounds good.

**Weaknesses:**

1. The method is not novel for the current stage. The main contributions of the paper include a random matrix projection and a LORA-based structure.
2. The author might need to justify more about the combination of two projectors. I understand the proof the author proposed that they aim to random anchor projections  to preserved the original feature characteristics and LRP residual reduces intra-class variance. However, mathemtically, could it possible to simplify the equation 4 into a single LORA-based structure $h(x) =h_{RA}(x) + h_{LRP}(x) \approx h_{LORA}(x)$?

3. The contribution of each projector ($h_{RA}(x)$, $h_{LRP}(x)$) is unclear. How do the authors combine each projection, and what is the weight for each part? Does the weight impact the results, and what is the robustness of the weight selection? These might impact the accuracy gain.

**Questions:**

1. What is RA-DL in Table 2? It only appears in Table 2 and without explanation.

**Details Of Ethics Concerns:**

This study involves the analysis of human data, but the data are open source.

---

> ### Author Response · Authors · 2025-11-21
> **[3-1] Response to Reviewer zaiN**
>
> We sincerely thank Reviewer zaiN for the thoughtful and constructive feedback. We also appreciate your recognition of the comprehensive experiments of our paper. Below, we address your suggestions and concerns point by point.
>
> > **Concern 1:** The method is not novel for the current stage. The main contributions of the paper include a random matrix projection and a LoRA-based structure.
>
> **Response:** We respectfully note that the reviewer may have misunderstood the core focus and contribution of this work. The **primary novelty** of RA-LDL is not the proposal of a new random matrix or a LoRA-like structure, but the **repositioning** of previously underutilized representation calibration techniques as effective and principled solutions for domain-specific continual learning in medical imaging. Rather than following the trend of increasingly complex PTM based CIL designs, RA-LDL highlights a minimalist, theoretically grounded and scalable alternative that is grounded in empirical observations specific to medical data.
>
> While some components have been independently explored in other tasks or in general domain CIL, their integration and complementary roles have not been systematically analyzed in the context of medical CIL. As stated in the abstract and introduction, our paper provides, to our knowledge, the first comprehensive analysis of their synergy and the first demonstration that such lightweight components are particularly effective for medical CIL. Many of these techniques were historically overlooked because they produce only modest gains in general-domain benchmarks, which led the community toward increasingly complex prompt-based, multi-adapter-based, and mixture-based designs. However, our extensive experiments reveal that these complex methods often become **unstable** in medical imaging, which exhibits low intra- and inter-class variability and strong inter-domain shifts. These properties amplify representation collapse and class correlation, making complex strategies brittle. In contrast, RA-LDL shows that simple, carefully composed calibration modules, when paired with strong PTM features, can achieve **dominant performance** in medical CIL, often rivaling or surpassing more sophisticated designs. We believe this finding is not only surprising but also **practically valuable**. It suggests that these lightweight strategies remain underutilized in the field, and their success in medical settings deserves more systematic attention.
>
> We further emphasize that medical CIL has not yet undergone the paradigm shift that has recently occurred in general-domain CIL. The field is still at an early stage, and PTM-based medical CIL remains largely unexplored. Our work represents an **early and pioneering** effort that identifies a promising and theoretically grounded direction. We believe RA-LDL can serve as a foundation for future research, either through deeper exploration of minimalist designs or through integration into broader medical AI frameworks.

---

> > ### Author Response · Authors · 2025-11-21
> > **[3-2] Response to Reviewer zaiN**
> >
> > > **Concern 2:** The author might need to justify more about the combination of two projectors. I understand the proof the author proposed that they aim to random anchor projections to preserved the original feature characteristics and LRP residual reduces intra-class variance. However, mathematically, could it possible to simplify the equation 4 into a single LORA-based structure $h(x)=h_{R A}(x)+h_{L R P}(x) \approx h_{\text {LoRA }}(x)$.
> >
> > **Response:** We thank the reviewer for the insightful question. LoRA can be an alternative to PEFT methods (e.g., VPT, SSF, and ViT-Adapter) as introduced in Appendix B, yet, regarding whether Eq. (4) can be simplified into a single LoRA-like structure, we believe this equivalence does not hold for two fundamental reasons.
> >
> > **(1) The Random Anchor (RA) module is not a LoRA-style low-rank adaptation of pretrained weights.** LoRA is typically designed for parameter-efficient fine-tuning (PEFT) of pretrained weight matrices, where a learnable low-rank update is added to an existing weight. In contrast, our RA is a **fixed random projection basis**. Its purpose is to embed features into a randomized higher-dimensional space to enhance linear separability without any training (Proof in Appendix H.2). RA does not update or adapt any pretrained weights, and therefore does not fit the underlying formulation or intent of LoRA.
> >
> > **(2) The Low-Rank Projection (LRP) operates as a residual correction on top of RA, rather than a low-rank update to the same weight matrix.** LRP is designed to calibrate residual distortions that may arise from domain shifts after the RA mapping (Proof in Appendix I). **Importantly**, LRP acts **in the output subspace of RA**, not on a shared pretrained weight matrix. Thus, its function is complementary to RA and structurally different from LoRA, which requires a single base weight undergoing a low-rank increment.
> >
> > Given these distinctions, this equivalence might not be appropriate. LoRA assumes a unified parameterization of the form $W+BA$, whereas our RA+LRP design follows a two-stage mechanism of (i) fixed random projection for stable anchoring and improved separability, and (ii) low-rank residual correction on the RA output space for robustness and domain-shift adaptation. These two modules operate in different spaces and pursue different objectives. In summary, RA and LRP are intentionally complementary for addressing the stability-plasticity trade-off in medical CIL, and might not be reduced to a single LoRA-like structure.
> >
> >
> > > **Concern 3:** The contribution of each projector $(h_{RA}(x), h_{LRP}(x)$ is unclear. How do the authors combine each projection, and what is the weight for each part? Does the weight impact the results, and what is the robustness of the weight selection? These might impact the accuracy gain.
> >
> > **Response:** We thank the reviewer for the question. We clarify that RA and LRP are not two losses that require manual weighting but two complementary feature-transformation pathways combined through a residual formulation. Because LRP is a learnable module rather than a manually weighted objective, the optimizer naturally determines its contribution relative to RA, and therefore no explicit hyperparameter is needed.
> >
> > > **Concern 4:** What is RA-DL in Table 2? It only appears in Table 2 and without explanation.
> >
> > **Response:** We thank the reviewer for pointing out the missing clarification. RA-DL refers to the variant of RA-LDL without LRP, included to isolate the contribution of the low-rank component in our ablation study. We have added this explanation for clarity.
> >
> > We sincerely thank Reviewer zaiN again for the insightful comments and careful reading. We are very happy to further discuss any remaining questions or technical details, and we **genuinely welcome continued active dialogue** to improve the clarity and impact of this work.

---

> > > ### Comment · Reviewer_zaiN · 2025-11-25
> > >
> > > I appreciate the authors giving me a complete explanation of my questions. I also carefully read our reviewers’ questions and the responses from the authors. As my first reviewer mentioned, I give credit to the authors for their application, experiments, and proof contributions to the continual learning community. During the rebuttal, for concern 4, I believe the authors’ explanation is great.
> > >
> > > However, I still feel the authors cannot address my concerns on structural novelty and the contribution of each component. This is similar to reviewer wD7K, W1. For example, the authors do not provide a clear ablation study or visualization to demonstrate RA and LRP contributions.

---

> > > > ### Author Response · Authors · 2025-11-28
> > > > **Additional Response to Reviewer ZaiN**
> > > >
> > > > We sincerely appreciate your positive recognition of our application-driven motivation, extensive experiments, and theoretical contributions to the continual learning community. We also thank you for the follow-up clarification of your concern.
> > > > - Regarding the concerns raised earlier, your comments focused on whether RA and LRP were essentially equivalent to LoRA. In our rebuttal, we explained in detail why this is not the case: their objectives, operating principles, and roles differ fundamentally. Our design is not a LoRA-style PEFT module, but rather a representation calibration mechanism that enables more effective analytical decorrelated learning, with complementary functions between the frozen random anchors and the single-session low-rank residual.
> > > > - We have also addressed Reviewer wD7K’s W1 in depth, which directly aligns with your structural-novelty concern. As summarized there, the novelty of RA-LDL lies in repositioning and systematizing previously underexplored representation calibration techniques into a minimalist, theoretically grounded, and empirically validated pipeline specifically for medical CIL. This perspective is distinct from prior PTM-based CIL works that rely on increasingly complex architectural modules, and RA-LDL serves as an early and pioneering exploration in PTM-based medical CIL.
> > > > - With respect to the contribution of each component, we apologize if our earlier discussion appeared incomplete, as you had not explicitly requested quantitative ablations. Nevertheless, comprehensive evidence is **provided in our response to Reviewer wD7K’s Concern 2**, including detailed ablation studies and sensitivity analyses that separately examine RA, LRP, and the analytical classifier. These results, combined with our theoretical analysis, clearly demonstrate the complementary value of each component.
> > > > - Taken together, we believe we have fully addressed both your initial and follow-up concerns. We respectfully hope that, given the strengthened clarifications, the additional ablation evidence, and Reviewer wD7K’s **positive** evaluation (which directly speaks to your current concerns), you may consider reassessing your score if our responses have resolved your reservations.
> > > >
> > > > We sincerely appreciate your time and constructive feedback.

---

### Official Review · Reviewer_XHJg · 2025-10-31

**Soundness:** 3
**Presentation:** 3
**Contribution:** 3
**Rating:** 4
**Confidence:** 4

**Summary:**

The paper addresses class-incremental learning (CIL) in medical image classification, where models must learn new disease categories over time without forgetting previously learned ones. The authors argue that existing PTM-based CIL methods—often complex and designed for natural images—fail in medical settings due to low intra-class variability and high inter-domain shifts (e.g., from different scanners or protocols). In response, they propose RA-LDL, a minimalist, representation-based pipeline with three components:
(a) optional one-time ViT-Adapter tuning for domain adaptation,
(b) feature calibration via a frozen Random Anchor (RA) projection and a single-session-trained Low-Rank Projection (LRP), and
(c) an analytical, closed-form decorrelated classifier based on ridge regression.
RA-LDL requires only one training session, avoids replay or complex routing, and achieves strong performance across four medical CIL benchmarks, often approaching joint-training upper bounds. The method is shown to work robustly with both general-domain and medical-specific pre-trained models (PTMs).

**Strengths:**

Strengths:
1.The RA-LDL approach is innovative in its simplicity. While other methods for class-incremental learning in medical imaging often rely on complex architectures, this paper proposes a lighter solution that offers competitive performance, especially under domain shift conditions. It effectively merges generalization and adaptability without extensive tuning.
2.By focusing on feature recalibration (using random anchors and low-rank projections), RA-LDL addresses low intra-class variability and high inter-domain shifts directly and effectively. The method shows solid promise in medical domains like COVID-19 diagnostics, blood cell analysis, and skin lesion classification.
3.It only requires one training session with minimal task-specific tuning, making it very practical for real-world medical applications where data is continually evolving.The approach works across various imaging modalities, making it applicable in diverse medical settings.

**Weaknesses:**

Weaknesses:
1.it could benefit from a more detailed examination of RA-LDL’s long-term stability and its ability to mitigate forgetting. Specifically, investigating the model's ability to retain previously learned classes as more tasks are introduced would strengthen the paper’s argument.
2.The “one training session” claim is slightly misleading: while only the first session trains the LRP, the classifier is updated incrementally using accumulated statistics. This is still efficient, but the phrasing could confuse readers expecting fully frozen inference.
3.How much memory is required to store G and Cp? For large class sets or high-dimensional features, the Gram matrix (d₁×d₁) could become costly. Please report memory usage or discuss scalability.
4.Why not compare to recent replay-free methods? While Table 2 includes some, a more direct comparison to non-PTM-based CIL would strengthen the claim that PTMs + RA-LDL are uniquely effective.

**Questions:**

Questions:
1.The authors show that general-domain PTMs, like ViT-B/16-IN21K, outperform medical-specific models such as UniMedCLIP or RAD-DINO. However, I have a question whether the improvements are more due to the general-domain PTM itself rather than the RA-LDL recalibration technique. The paper could further clarify how RA-LDL benefits from this general-domain foundation and whether it can achieve similar results when applied to domain-specific PTMs.
2.While the manuscript evaluates the model’s robustness to task ordering, further analysis on how RA-LDL performs in highly unstructured or unpredictable sequences of class introduction (i.e., non-sequential class introduction) could be informative. For example, testing RA-LDL under random or shuffled class orders to assess how well it maintains generalizability when the sequence of tasks is not controlled or predictable.
3.Are there scenarios where RA-LDL underperforms? For example, in highly imbalanced tasks (e.g., Skin8), does decorrelation hurt minority classes?

---

> ### Author Response · Authors · 2025-11-21
> **[2-1] Response to Reviewer XHJg**
>
> We sincerely thank Reviewer XHJg for the constructive and comprehensive feedback. We are pleased that you found our method innovative in its simplicity, as well as practical and broadly applicable to real-world medical scenarios. Below, we address your suggestions and concerns point by point.
>
> > **Concern 1:** The paper could benefit from a more detailed examination of RA-LDL’s long-term stability and its ability to mitigate forgetting. Specifically, investigating the model's ability to retain previously learned classes as more tasks are introduced would strengthen the paper’s argument.
>
> **Response:** We appreciate the reviewer’s interest in the long-term stability of RA-LDL and its ability to mitigate forgetting. We would like to clarify that our paper **already includes an initial examination** of this aspect, though it may have been inadvertently overlooked. Specifically, in the Experimental Analysis section, we provide a dedicated subsection titled “Robustness to the Number of Tasks”, where we intentionally increase the number of incremental tasks on the MedMNIST-Sub dataset (which contains 36 classes and is therefore suitable for finer task partitioning). As shown in Appendix G.5, when increasing the task number from 4 to 8, RA-LDL maintains highly stable performance, where $Acc_\text{Avg}$ changes only from 84.79% to 86.71% and $Acc_\text{Last}$ from 70.60% to 70.80%, demonstrating strong stability as more tasks accumulate.
>
> We also note that most medical datasets contain far fewer classes than general-domain benchmarks, making it inherently challenging to construct extremely long incremental sequences within a single dataset. To address this limitation, we further created a cross-dataset long-term continual learning benchmark, where the original intra-dataset task splits (as shown in Table 1 in the manuscript) are preserved, and additional tasks are introduced across different datasets. This setting provides a much more demanding and realistic long-horizon evaluation of knowledge retention and forgetting. The results below (**now added to the Appendix**) consistently show that RA-LDL sustains high performance even under this extended and heterogeneous learning trajectory, confirming its robustness in long-term continual adaptation.
>
> Overall, both the intra-dataset and cross-dataset analyses jointly validate that RA-LDL exhibits strong long-term stability and effective mitigation of forgetting, even as task sequences become substantially longer and more challenging.
>
> **Table**: Performance comparison on cross-dataset long-term continual learning benchmark (Order: Skin8 - COVID - MedMNIST - BLOOD), following original intra-dataset task splits in Table 1 (of the manuscript).
> | Framework      | $Acc_\text{Avg}$ | $Acc_\text{Last}$ |
> |----------------|---------|---------|
> | Joint Training (upper bound) |-|71.65|
> | SimpleCIL |57.50|55.41|
> | ADAM-Adapter |63.18|55.52|
> | SSIAT |53.97|28.41|
> | MOS  |70.66|62.05|
> | **RA-LDL (ours)**  |**74.56**|**71.52**|
>
> > **Concern 2:** The “one training session” claim is slightly misleading: while only the first session trains the LRP, the classifier is updated incrementally using accumulated statistics. This is still efficient, but the phrasing could confuse readers expecting fully frozen inference.
>
> **Response:** We appreciate the reviewer’s observation. Our intention with the phrase “one training session” was to emphasize that all learnable parameters (the ViT-adapter and the LRP) are optimized only in the first task, and no further gradient-based training is performed afterward. We agree that the wording could inadvertently imply that the entire system, including the classifier, remains fully frozen. To avoid confusion, we will rephrase the manuscript to clarify that subsequent sessions involve only closed-form analytic updates to the classifier using accumulated statistics: RA-LDL requires gradient-based optimization only in the first session; subsequent tasks rely solely on efficient analytic classifier updates based on recursively accumulated statistics, without any further network training.

---

> > ### Author Response · Authors · 2025-11-21
> > **[2-2] Response to Reviewer XHJg**
> >
> > > **Concern 3:** How much memory is required to store G and Cp? For large class sets or high-dimensional features, the Gram matrix (d₁×d₁) could become costly. Please report memory usage or discuss scalability.
> >
> > **Response:** We thank the reviewer for raising this practical concern. RA-LDL stores only two recursively updated statistics, the Gram matrix $G \in \mathbb{R}^{d_1 \times d_1}$ and the class-accumulated feature matrix $C_{p} \in \mathbb{R}^{d_1 \times |Y_t|}$. This design avoids retaining any samples or feature buffers, and the memory requirement is $\mathcal{O}\left(d_1^2+d_1\left|\mathcal{Y}_t\right|\right)$, independent of dataset size or the number of tasks. With our configuration $d_1 = 5 d_0$ (for example $d_0 = 768$ which gives $d_1 = 3840$ ), the Gram matrix $G$ requires approximately 56 MB in FP32 and $C_p$ adds less than 2 MB even for dozens of classes. This is significantly lighter than replay buffers or PTM-based methods that maintain prompt pools, adapter banks, or multiple model copies. Since only the classifier $W$ grows with the number of classes while $G_t$ remains fixed in dimension, RA-LDL is both memory-efficient and scalable for medical CIL. In addition, on an A100 GPU, the end-to-end inference time of RA-LDL is comparable to SimpleCIL (which uses a frozen PTM and a prototype-based classifier), with average latency of 2.53 ms vs. 2.49 ms per image on the Skin8 dataset. The ViT-adapter and LRP together introduce only 1.19M trainable parameters, and they are trained in a single initial session.
> >
> >
> > > **Concern 4:** Why not compare to recent replay-free methods? While Table 2 includes some, a more direct comparison to non-PTM-based CIL would strengthen the claim that PTMs + RA-LDL are uniquely effective.
> >
> > **Response:** We appreciate the reviewer’s suggestion. In our preliminary experiments, replay-free conventional CIL approaches such as LwF and EWC performed **suboptimally** in our heterogeneous medical setting. LwF suffered from severe forgetting despite its distillation-based mechanism, and EWC’s Fisher-information weighting was insufficient to constrain parameter drift under severe distribution shifts (Skin8 dataset: $Acc_\text{Last}$ of 11.35% for LwF and 16.74% for EWC). These results indicate that their regularization-based mechanisms are inadequate for constraining parameter drift under the strong inter-domain shifts and compressed inter- and intra-class variability characteristic of medical imaging.
> >
> > In contrast, most conventional CIL methods, such as FOSTER, iCaRL, DER, and ACL, rely on a replay buffer, which has been shown to be essential for maintaining performance in the absence of pretrained representations. When the buffer is removed, these approaches exhibit a substantial drop in accuracy, confirming that replay is a necessary component for stability in traditional CIL. Replay-free continual learning becomes practically achievable only in the PTM era, where large-scale pretrained representations provide strong generalization and intrinsic stability. Our RA-LDL framework builds on this paradigm, showing that effective replay-free adaptation can be achieved when leveraging PTMs, a property that conventional CIL methods fundamentally lack.

---

> > > ### Author Response · Authors · 2025-11-21
> > > **[2-3] Response to Reviewer XHJg**
> > >
> > > > **Concern 5:** The authors show that general-domain PTMs, like ViT-B/16-IN21K, outperform medical-specific models such as UniMedCLIP or RAD-DINO. However, I have a question whether the improvements are more due to the general-domain PTM itself rather than the RA-LDL recalibration technique. The paper could further clarify how RA-LDL benefits from this general-domain foundation and whether it can achieve similar results when applied to domain-specific PTMs.
> > >
> > > **Response:** We appreciate the reviewer’s thoughtful question regarding whether the improvements arise primarily from the general-domain PTM itself. It is important to emphasize that all PTM-based CIL methods fundamentally depend on the quality of the pretrained feature extractor. No continual adaptation strategy, including ours, can fully compensate for an intrinsically weak or poorly aligned PTM, and the representational strength of the backbone necessarily constrains the achievable performance ceiling in any PTM-based continual learning pipeline.
> > >
> > > However, as shown in Table 3 and Section 5.2, RA-LDL consistently delivers **clear improvements within each PTM family**, whether general-domain or medical-specific, when compared to methods using the same backbone. This confirms that our gains are not simply inherited from stronger general-domain features, but stem from RA-LDL’s recalibration mechanism itself. At the same time, the comparison highlights a complementary pattern: general-domain PTMs tend to offer more well-generalized cross-domain features, while medical-specific PTMs provide domain-oriented priors. RA-LDL is effective in both regimes, whereas existing CIL baselines typically excel in only one. These findings highlight RA-LDL’s robustness and broad applicability across diverse pretrained models.
> > >
> > > In fact, we view PTMs and RA-LDL as mutually reinforcing. As medical-specific PTMs continue to improve, RA-LDL can further unlock their potential, making their co-evolution a natural direction for future work.
> > >
> > >
> > > > **Concern 6:** While the manuscript evaluates the model’s robustness to task ordering, further analysis on how RA-LDL performs in highly unstructured or unpredictable sequences of class introduction (i.e., non-sequential class introduction) could be informative. For example, testing RA-LDL under random or shuffled class orders to assess how well it maintains generalizability when the sequence of tasks is not controlled or predictable.
> > >
> > > **Response:** We thank the reviewer for the suggestion. We would like to clarify that **our experimental setup does not rely on any fixed or semantically meaningful class sequence**. Following common practice in PTM-based CIL, all experiments use a **predefined randomly shuffled task/class order**, ensuring that the sequence is inherently unstructured and not based on any anatomical or semantic grouping.
> > >
> > > To further test robustness under additional unpredictable sequences, we additionally evaluate RA-LDL under a reversed version of this shuffled order. As shown in Table 4, RA-LDL exhibits only minimal variation across these random and reversed sequences, confirming that its generalizability is **not sensitive to how classes are introduced**, even when the order is entirely arbitrary and uncontrolled. We have rephrased the manuscript accordingly to make this setup clearer.
> > >
> > >
> > > > **Concern 7:** Are there scenarios where RA-LDL underperforms? For example, in highly imbalanced tasks (e.g., Skin8), does decorrelation hurt minority classes?
> > >
> > > **Response:** We thank the reviewer for raising this important question. Empirically, RA-LDL consistently outperforms all baselines across the four medical datasets, including the highly imbalanced ones such as Skin8. Interestingly, these challenging datasets are where the analytical decorrelation mechanism provides **larger relative gains**, suggesting that reducing prototype correlation and mitigating feature collapse particularly benefits minority classes under severe imbalance. Moreover, RA-LDL’s performance on these datasets approaches the joint-training upper bound, indicating that decorrelation does not harm minority classes; instead, it improves feature alignment and stabilizes representation learning under imbalance.
> > >
> > > A scenario where RA-LDL may underperform is when the PTM itself provides weak or poorly aligned representations (e.g., some modality-specific PTMs such as RAD-DINO in Table 3). However, this limitation is shared by all PTM-based CIL approaches. The attainable ceiling is fundamentally constrained by the backbone’s representational strength. We believe ongoing progress in both general-domain and medical-specific PTMs will continue to enhance RA-LDL’s effectiveness.
> > >
> > > We will extend our evaluation to segmentation and diagnostic tasks in future work to further assess its generalizability and may provide deeper insights into edge-case performance.

---

> > > > ### Author Response · Authors · 2025-11-21
> > > > **[2-4] Response to Reviewer XHJg**
> > > >
> > > > We sincerely thank Reviewer XHJg again for the careful suggestion. We are very happy to further clarify any remaining questions, and we genuinely **welcome active discussion** to improve the clarity and impact of our work.

---

> > > > > ### Author Response · Authors · 2025-11-28
> > > > > **Follow-up Discussion**
> > > > >
> > > > > Again, we sincerely thank Reviewer XHJg for the careful suggestion. We respectfully hope the reviewer check our responses and consider updating the score if our responses have satisfactorily addressed the earlier points.

---

### Official Review · Reviewer_wD7K · 2025-11-01

**Soundness:** 4
**Presentation:** 3
**Contribution:** 3
**Rating:** 6
**Confidence:** 3

**Summary:**

The paper introduces a minimalist approach for class-incremental learning (CIL) in medical image classification called Random Anchors with Low-rank Decorrelated Learning (RA-LDL). The motivation is to address the unique challenges of medical CIL, which involves low intra-class variability and high inter-domain shifts, where complex pre-trained model (PTM)-based continual learning methods from the general domain often fail. RA-LDL combines a feature extraction method, lightweight feature calibration via a frozen random anchor projection and a single-session-trained Low-Rank Projection, and a closed-form analytical de-correlated classifier based on ridge regression.

**Strengths:**

The empirical evaluation is thorough, benchmarked across four realistic medical datasets as well as standard general-domain CIL benchmarks, and ablation studies clearly show the contribution of each component.

The work points out an overlooked strategy to CIL that offers better scaling properties/simplicity. It advances methodological understanding by showing that domain-specific challenges in medical continual learning are best addressed with statistical feature recalibration, not greater architectural/training complexity, which is an applicable insight to other fields with similar data properties.

Theoretical justifications for each step (random anchor, LRP, analytical de-correlated classifier) are provided, including proofs of distance and covariance preservation, variance reduction, and prototype de-correlation.

**Weaknesses:**

While the synergy and systematic evaluation of random anchors, low-rank projections, and de-correlated classifiers are novel for medical CIL, many components are incremental adaptations or well-established in general-domain CIL (Zhuang et al., 2022, McDonnell et al., 2024). More in-depth, quantitative ablation and comparison with these specific prior works could clarify the genuine novelty and highlight improvements beyond re-combination. Some discussion in appendix, but would be nice to see more quantitative evidence.

**Questions:**

See above.

---

> ### Author Response · Authors · 2025-11-21
> **[1-1] Response to Reviewer wD7K**
>
> We sincerely thank Reviewer wD7K for the constructive and comprehensive feedback. We are pleased that you found our paper to provide a thorough empirical evaluation, valuable methodological insight into scalable medical CIL, and strong theoretical justifications for each design. Below, we address your suggestions and concerns point by point.
>
> > **Concern 1:** While the synergy and systematic evaluation of random anchors, low-rank projections, and decorrelated classifiers are novel for medical CIL, many components are incremental adaptations or well-established in general-domain CIL.
>
> **Response:** We thank the reviewer for this comment. We respectfully clarify that the **primary novelty** of RA-LDL does **not** lie in inventing entirely new primitives, but the **repositioning** of previously underutilized representation calibration techniques as effective and principled solutions for domain-specific continual learning in medical imaging. Rather than following the trend of increasingly complex PTM-based CIL designs, RA-LDL highlights a minimalist, theoretically grounded and scalable alternative that is grounded in empirical observations specific to medical data.
>
> While some components have been independently explored in other tasks or in general-domain CIL, their integration and complementary roles have not been theoretically and systematically analyzed in the context of medical CIL. As stated in the abstract and introduction, our paper provides, to our knowledge, the first comprehensive analysis of their synergy and the first **demonstration** that such lightweight components are particularly effective for medical CIL. Some techniques were **historically overlooked** because they produce **only modest gains in general-domain benchmarks**, which led the community toward increasingly complex prompt-based, multi-adapter-based, and mixture-based designs. However, our extensive experiments reveal that these complex methods often become **unstable** in medical imaging, which exhibits low intra- and inter-class variability and strong inter-domain shifts. These properties amplify representation collapse and class correlation, making complex strategies brittle. In contrast, RA-LDL shows that simple, carefully composed calibration modules, when paired with strong PTM features, can achieve **dominant performance** in medical CIL, often rivaling or surpassing more sophisticated designs. We believe this finding is not only surprising but also **practically valuable**. It suggests that these lightweight strategies remain **underutilized** in the field, and their success in medical settings deserves more systematic attention.
>
> We further emphasize that medical CIL has not yet undergone the paradigm shift that has recently occurred in general-domain CIL. The field is still at an early stage, and PTM-based medical CIL remains largely unexplored. Our work represents **an early and pioneering effort** that identifies a promising and theoretically grounded direction. We believe RA-LDL can serve as a foundation for future research, either through deeper exploration of minimalist designs or through integration into broader medical AI frameworks.

---

> > ### Author Response · Authors · 2025-11-21
> > **[1-2] Response to Reviewer wD7K**
> >
> > > **Concern 2:** More in-depth, quantitative ablation and comparison with specific prior works could clarify the genuine novelty and highlight improvements beyond re-combination. Some discussion in the appendix, but would be nice to see more quantitative evidence.
> >
> > **Response:** We thank the reviewer for raising this point. We would like to clarify that our paper intentionally adopts a step-by-step, analysis-driven presentation style, where each component of RA-LDL is introduced together with its theoretical justification and immediate empirical support, rather than separating all ablations into a standalone section. This design follows the statement made at the beginning of Sec. 3.3:
> > > *“Since we will progressively combine theoretical analysis and empirical evidence to construct the RA-LDL pipeline step by step…”*
> >
> > As a result, several quantitative ablations appear **within the methodology section and corresponding appendix** and **may have been overlooked**. Concretely:
> > - Analytical Decorrelated Classifier (ADC): We provide both the formal decorrelation proof (Appendix F) and direct quantitative evidence in Table 5. For example, ADC alone boosts $Acc_\text{Last}$ on MedMNIST-Sub from 50.45 to 66.44 (Zhuang et al., 2022) (For reference, the entire RA-LDL is 70.60).
> > - Random Anchors (RA): When introducing RA, we conduct projection-dimension ablations (Appendix G.2) and random-matrix initialization studies (Appendix G.1), showing how RA affects separability and stability, far beyond simply adopting (McDonnell et al., 2024). These experiments also motivate the need for LRP.
> > - Low-Rank Projection (LRP): We include a rank-sensitivity study (Appendix G.3) and provide formal variance-reduction and margin-enlargement analyses (Appendix I), showing that LRP contributes non-trivially to robustness under domain shift.
> > - Full RA-LDL Synergy: Table 2 (last four rows) provides a classical overall ablation. The improvements across all medical benchmarks directly demonstrate that RA-LDL’s performance cannot be explained by simple component reuse.
> >
> > Additionally, we include macro-level robustness studies, i.e., PTM variability (Table 3), task-order robustness (Table 4), and task-number scalability (Appendix G.5), to ensure that each component generalizes across realistic continual-learning conditions.
> >
> > In summary, although we did not use the traditional “Method” and “Experiments” section separation, the paper contains rich, fine-grained quantitative ablations tightly integrated into the pipeline construction. We appreciate the reviewer’s feedback and have clarified these placements to make the evidence more visible.
> >
> > We sincerely thank Reviewer wD7K again for the careful suggestion. We are very happy to further clarify any remaining questions, and we genuinely **welcome active discussion** to improve the clarity and impact of our work.

---

### Author Response · Authors · 2025-12-02
**Summary Remark to ACs**

We thank all reviewers for their careful reading, constructive feedback, and positive recognition of our paper’s strengths, including comprehensive experiments, theoretically grounded design, and practical relevance for scalable medical continual learning.

Across the rebuttal, we have **directly responded to all major concerns** raised by the reviewers:

**1. Clarification of overlooked evidence and misinterpretations:** Several concerns stemmed from overlooked results already present in the manuscript/appendix, including:
- fine-grained ablations (RA, LRP, analytical decorrelation, rank sensitivity, projection dimension),
- robustness studies (task number, task order, PTM variation), and
- long-term stability evaluation (intra-dataset and cross-dataset (new results) task growth).

We cited the exact sections, tables, and appendices to make these contributions more visible.

**2. Structural novelty and positioning:** We clarified that our novelty and contribution lie in repositioning overlooked representation-calibration techniques into a minimalist, theoretically justified, and medically effective continual-learning pipeline, which is a perspective that has been undervalued and, to our knowledge, not systematically explored in medical CIL.

**3. Resolution of specific technical doubts:** We clarified reviewer misunderstandings around:
- RA vs. LoRA (fundamentally different objectives and parameterization),
- “one-training-session” meaning (gradient-based training only once; subsequent tasks use closed-form updates),
- memory/scalability of Gram matrices,
- random versus controlled task ordering,
- and whether gains are inherited solely from strong PTMs (they are not; our calibrations consistently improve within each PTM family).

Besides, we would like to respectfully highlight again that **all reviewers praised important aspects** of our work:
- Reviewer wD7K commended our “thorough **empirical evaluation, valuable methodological insight, and strong theoretical justifications**.”
- Reviewer XHJg found RA-LDL “**innovative in its simplicity as well as practical and broadly applicable to real-world medical scenarios.**”
- Reviewer zaiN positively recognized our “**application-driven motivation, comprehensive experiments, and proof contributions**,” even in the follow-up message.

Taken together, we believe the rebuttal **successfully clarifies misinterpretations, makes previously overlooked evidence explicit, and resolves core concerns** around **novelty, component contribution, and robustness**.

We sincerely hope these strengthened explanations will assist AC in making an informed recommendation, and we are happy to further discuss any remaining questions.

---

### Meta-Review · Area_Chair_RtAo · 2026-01-07

**Summary:**

The paper proposes an approach for class-incremental learning (CIL) in medical image classification.  The motivation is to tackle low intra-class variability and high inter-domain shifts where pre-trained model-based continual learning fails. The proposal combines feature extraction, feature calibration (based on a frozen random anchor projection), and a low-rank projection paired with a classifier.

The reviewers raised several concerns, and the authors provided answers to most of the issues, except the limited contributions given the reuse of existing components.  I consider most of the issues addressed and believe that the reviewers could have increased their scores.  However, given the limited contribution I also see the reviewers leaning towards a rejection as well.  Thus, I recommend an accept but wouldn't mind a reject either.


Strengths:
- Empirical evaluation is thorough (evaluation on 4 medical datasets and standard general domain benchmarks, and ablation studies that evaluate each component)
- The paper works on an overlooked strategy to CIL
- The proposal simplifies complex existing pipelines while providing a theoretical foundation
- The focus on feature recalibration addresses low intra-class variability and high inter-domain shifts
- The proposed method requires one training session with minimal task-specific tuning


Weaknesses:
- Many of the components used are well-established in the general-domain CIL.
- While the evaluation is thorough, the comparisons against existing approaches is missing
- The examination of the long-term stability to mitigate forgetting the learned classes could be more extensive
- The one training session claim must be revised and aligned with the actual procedure
- The components of the proposal were not thoroughly evaluated

**Reviewer Concerns:**

Reviewer wD7K had concerns about the incremental proposal, and the missing comparisons against existing prior works.

Reviewer XHJg is concerned about the long-term stability of the proposed method to forgetting, the one training session claim, the memory requirements to store the Gram matrix, the reason for the improvements being due to the general domain pre-trained models or the proposed recalibration is unclear, the impact of the task ordering is unclear as well.

Reviewer zaiN raised novelty concerns given the incremental contributions, the similarity between the proposal and a LoRA method, as well as evaluation of the contribution of each of the components.

**Reviewer Scores:**

Reviewer wD7K recommended a borderline accept given the thorough empirical evaluation despite the limited comparisons against other recent methods, and the incremental methodological contribution.  The authors replied that their contribution is the theoretical repositions of the existing body of work and the principled demonstration that simpler models are successful and underutilized.

Reviewer XHJg recommended a borderline reject.  The long-term stability issues were addressed by the authors' rebuttal where they pointed out that it is partially in the paper already, and longer sequences were added to the appendix. The reviewer also asked about the lack of comparisons against replay-free methods, to which the authors replied that they tested them but performed poorly. Despite the low rating, the strengths of the paper outweigh the raised weaknesses.

Reviewer zaiN recommended a borderline reject.  The authors replied about the contribution being the repositioning and simplification over the use of existing components, and explained that the random anchor and the low-rank projection while similar to a LoRA are different and have different roles.  The authors mentioned extensive experiments to evaluate these two components.  While the reviewer commented that their concerns were not addressed, the authors provided additional experiments and showed that the proposal worked.  However, the incremental contribution still remains.

---

### Decision · Program_Chairs · 2026-01-26

Accept (Poster)